# Disruption of polyhomeotic polymerization decreases nucleosome occupancy and alters genome accessibility

Adfar Amin[1],*, Sangram Kadam[2],*, Jakub Mieczkowski[3], Ikhlak Ahmed[4], Younus A Bhat[1], Fouziya Shah[1], Michael Y Tolstorukov[3], Robert E Kingston[3,5], Ranjith Padinhateeri[2], Ajazul H Wani[1]

**Chromatin attains its three-dimensional (3D) conformation by establishing contacts between different noncontiguous regions. Sterile Alpha Motif (SAM)–mediated polymerization of the polyhomeotic (PH) protein regulates subnuclear clustering of Polycomb Repressive Complex 1 (PRC1) and chromatin topology. The mutations that perturb the ability of the PH to polymerize, disrupt long-range chromatin contacts, alter Hox gene expression, and lead to developmental defects. To understand the underlying mechanism, we combined the experiments and theory to investigate the effect of this SAM domain mutation on nucleosome occupancy and accessibility on a genome wide scale. Our data show that disruption of PH polymerization because of SAM domain mutation decreases nucleosome occupancy and alters accessibility. Polymer simulations investigating the interplay between distant chromatin contacts and nucleosome occupancy, both of which are regulated by PH polymerization, suggest that nucleosome density increases when contacts between different regions of chromatin are established. Taken together, it appears that SAM domain–mediated PH polymerization biomechanically regulates the organization of chromatin at multiple scales from nucleosomes to chromosomes and we suggest that higher order organization can have a top–down causation effect on nucleosome occupancy.**

## Introduction

Chromatin within the cell nucleus is organized in a complex, nonrandom 3D conformation. A generic feature of chromatin folding, well accepted, is its hierarchical nature (1, 2, 3). The organizational complexity increases from nucleosomes to the formation of simple chromatin loops and topologically associating domains (TADs), which form because of preferential contacts within a genomic region as compared with neighboring regions. TADs of the same type aggregate and result in the formation of A and B compartments. This type of hierarchy seems to continue up to the scale of entire chromosome via formation of meta-TADs of increasing size (4, 5). These organizational features have been observed in different organisms and cell types, implying the existence of fundamental underlying principles governing the architecture of chromatin. 3D chromatin organization is linked to the regulation of chromatin-associated processes like gene expression, replication, and repair which occur at the nucleosome level (5, 6, 7, 8, 9), but, the mechanistic details of how higher order chromatin folding exerts its effects at the level of nucleosomes is not well understood.

3D organization of chromatin is shaped by biochemical and by biomechanical mechanisms. The polymeric nature of chromatin, nuclear confinement, and the nuclear lamina impart mechanical constraints which can influence the 3D folding of chromatin (10, 11, 12, 13, 14). 3D folding of chromatin is achieved by formation of contacts between different noncontiguous regions mediated by protein–protein interactions. Crosslinking density in the case of synthetic polymers has been shown to modulate various properties like stiffness, volume, temperature dependence, etc. (15, 16, 17). Another mechanical property affected by crosslinking is polymer chain dynamics, which decreases with increasing crosslinking (17). Chromatin, as a polymer (18) can also possess these properties. For example, the number and strength of chromatin contacts can influence the properties of the chromatin chain composed of nucleosomes.

Folding of chromatin is driven by many non-histone chromatin-associated proteins like CCCTC binding factor (CTCF), cohesin, Polycomb Group (PcG) proteins, etc. (19, 20, 21, 22, 23, 24). PcG proteins, conserved from *Drosophila* to humans, modulate chromatin organization either biochemically by modifying histones or biomechanically by physically constraining and compacting chromatin

[1]Department of Biotechnology, School of Biological Sciences, University of Kashmir, Srinagar, India    [2]Department of Biosciences and Bioengineering, IIT, Bombay, India    [3]Department of Molecular Biology, Massachusetts General Hospital, Boston, MA, USA    [4]CIRI, School of Biological Sciences, University of Kashmir, Srinagar, India    [5]Department of Genetics, Harvard Medical School, Boston, MA, USA

Correspondence: ahwani@kashmiruniversity.ac.in
Jakub Mieczkowski's present address is Medical University of Gdansk, International Research Agenda 3P Medicine Laboratory, Gdansk, Poland
Ikhlak Ahmed's present address is Department of Human Genetics, Sidra Medicine, Doha, Qatar
Michael Y Tolstorukov's present address is Bioinformatics and Data Science Group, Informatics and Analytics Department, Dana-Farber Cancer Institute, Boston, MA, USA
*Adfar Amin and Sangram Kadam contributed equally to this work

(23, 25, 26, 27, 28). Evidence for biomechanical basis of PcG protein-mediated folding of chromatin comes from in vitro and in vivo studies. In vitro, Polycomb repressive complex 1 (PRC1) or its subunits compact oligonucleosomes, an activity conserved across different species (29, 30, 31). In cells, PRC1 modulates the topology of chromatin by binding to the chromatin at specific sites and mediating contacts between noncontiguous regions of chromatin (32). This activity stems from the polymerization property of the SAM domain of PRC1 subunit, PH. The PH SAM domain polymerizes in a head-to-tail polymer via its mid-loop (ML) and end-helix (EL) motifs (33). Disruption of PH polymerization results in a decrease in long-range chromatin contacts over multiple genomic distances (32, 34). Knock down of PH induces decompaction of the Bithorax–Complex (*BX-C*) gene cluster in *Drosophila* when visualized by super-resolution imaging (35).

Specific mutations in ML or EL abolish the polymerization ability of PH, cause derepression of Hox genes, and give rise to skeletal defects in mice (32, 33, 34). SAM domain–mediated polymerization of PH results in the formation of nanoscale subnuclear clusters which get dissociated upon disruption of PH polymerization by specific mutations in ML (32, 34, 36). Hence, molecules of PH bound at specific sites along the chromatin seem to interact via their SAM domain, resulting in folding of the underlying chromatin fiber and maintenance of proper gene expression; but, whether the higher order folding of chromatin mediated by PH polymerization influences the properties of the underlying nucleosomes has not been studied.

Here, we investigated the effect of SAM domain–mediated polymerization of PH on occupancy and accessibility of nucleosomes on a genome-wide scale. Disruption of PH polymerization decreases nucleosome occupancy and alters accessibility. We used polymer modeling to simulate the interplay between chromatin contacts and the density of underlying nucleosomes, both of which are modulated by PH polymerization. These simulations suggest that the density of nucleosomes increases when distant chromatin contacts are established.

## Results

### PH polymerization influences nucleosome occupancy

SAM domain mediated PH–PH interaction mediates chromatin contacts and specific mutations which disrupt this interaction result in loss of long-range chromatin contacts (32), but the effect of this mutation on occupancy of nucleosomes has not been studied. We used MNase-titration-seq (37) to accurately determine the nucleosome occupancy on a genome-wide scale. We choose MNase-titration-seq because nucleosomes are known to have differing sensitivity to internal cleavage by MNase, so accurately measuring nucleosome occupancy requires integrating across multiple MNase concentrations. We measured occupancy of nucleosomes at five different concentrations of MNase in three different cell lines: *Drosophila* S2 cells, S2 cells stably over expressing either wild type PH (PH-WT) or a SAM polymerization defective mutant of PH (PH-ML) (Fig S1) under an inducible promoter. Two independent

experiments, each having five MNase-seq data sets were carried. Therefore, each occupancy value shown is an average of 10 different independent MNase-seq experiments for each cell line. PH is endogenously encoded by *ph-p* and *ph-d* genes. Given the duplicated nature, its perturbation by CRISPR-Cas9 technology is very difficult. PH polymerizes in a head-to-tail manner via its ML and EL motifs. PH-ML is a dominant negative mutation (32, 34), therefore, once incorporated, it will prevent further polymerization even in the presence of wild type PH. Given these aspects, an overexpression system under an inducible promoter was used. To dissect whether the observed effects are not just because of overexpression, we also analyzed PH-WT overexpressing cells, but observed opposing effects on nucleosome occupancy.

To determine whether disruption of PH polymerization alters nucleosome occupancy, the entire genome was binned into 300-bp non-overlapping bins and differences in averaged nucleosome occupancy were computed by subtracting occupancy obtained in S2 cells from nucleosome occupancy values obtained from either PH-ML- or PH-WT–expressing cells. A clear decrease in nucleosome occupancy is observed in PH-ML–expressing cell as compared with S2 cells (Fig 1A). The values underlying the heatmap corresponding to PH-ML–S2 are mostly below zero. However, an increase in nucleosome occupancy is observed in PH-WT–expressing cells as compared with PH-ML–expressing cells (Fig 1A). The same result is also evident from correlation and clustering of nucleosome occupancy values obtained from three cell lines and two biological replicates. Nucleosome occupancy of PH-ML–expressing cells is less similar to nucleosome occupancy of S2 cells and the similarity further deceases upon comparison with PH-WT–expressing cells (Fig 1B). Furthermore, correlation between the differences in nucleosome occupancy from replicates also shows that replicates for a particular comparison are more similar to each other than replicates for other comparisons (Fig S2A). Parallel comparison of PH-ML–expressing cells with S2 cells and PH-WT–expressing cells with S2 cells was carried to find whether the effects observed do not arise merely because of overexpression of PH but because of mutation in the ML motif. *Drosophila* S2 cells served as an unperturbed control.

Nucleosome occupancy profiles aligned by their TSS (Fig S2B) show (i) a strong +1 nucleosome, (ii) a nucleosome-depleted region at the TSS, and (iii) regular phasing of nucleosomes. In comparison to S2 cells, cells expressing PH-ML show a decrease in nucleosome occupancy at lower MNase concentrations (Fig S2B). The decrease in occupancy in case of PH-ML is also observed upon comparison of averaged occupancies (obtained from five different MNase concentrations). We also analyzed change in nucleosome occupancy in cells overexpressing PH-WT protein in comparison to S2 cells; PH-WT expressing cells show a moderate increase in nucleosome occupancy at lower MNase concentrations (Fig S2B). These results suggest that upon disruption of PH polymerization, which decreases noncontiguous chromatin contacts, there is a significant decrease in nucleosome occupancy.

### Change in nucleosome occupancy around PH-binding sites

Upon observation of differences in global comparisons, we investigated the relationship between nucleosome occupancy and

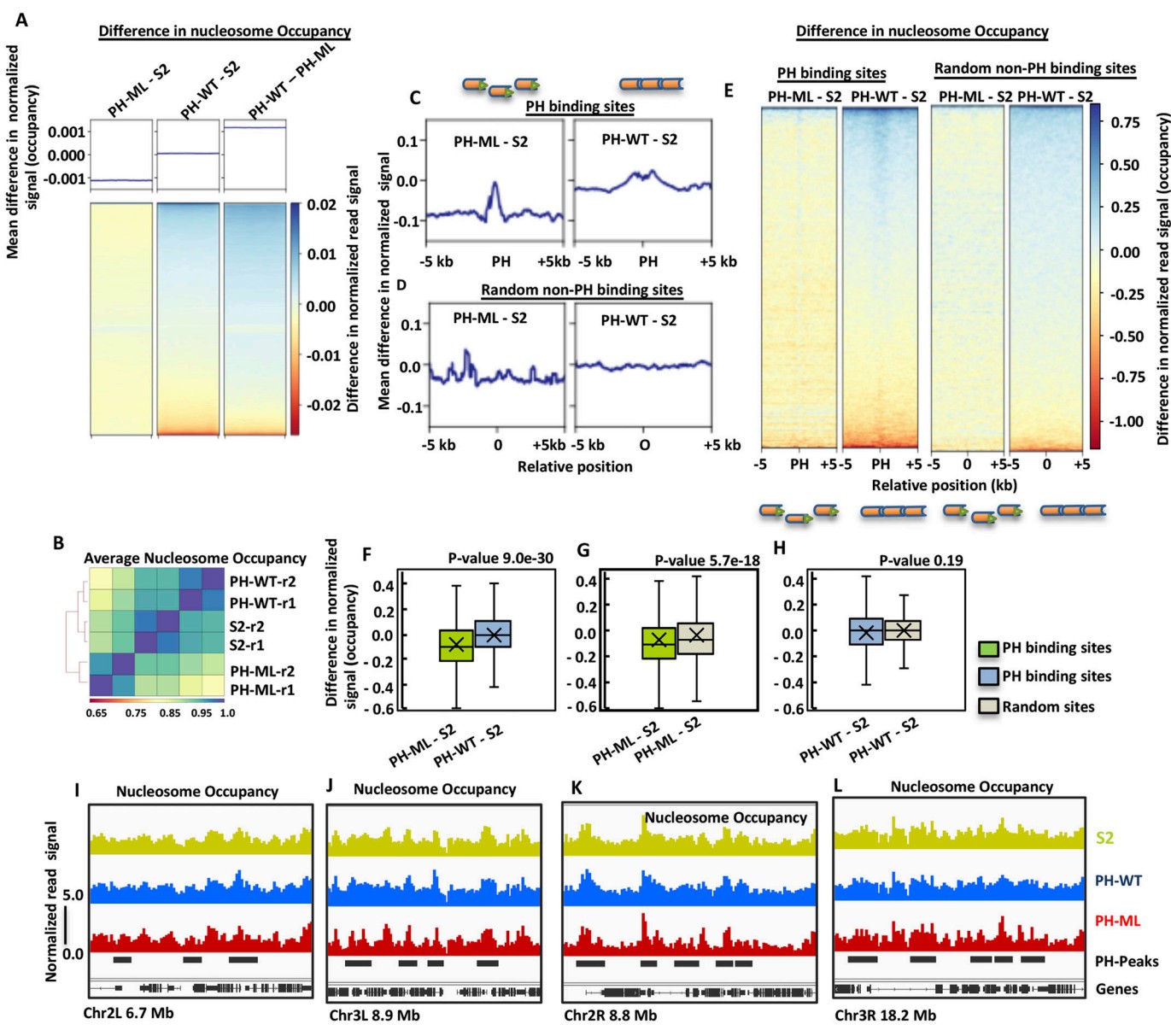

**Figure 1. Disruption of PH polymerization changes nucleosome occupancy.**
**(A)** Averaged genome-wide difference in nucleosome occupancy computed in 300-bp bins between three cell lines. The differences were obtained by subtracting nucleosome occupancy of the corresponding genomic bins. Heatmaps show genome-wide differences in average nucleosome occupancy in nonoverlapping 300-bp bins across the entire genome in descending order. **(B)** Correlation between averaged nucleosome occupancy obtained from different biological replicates and from three different cell lines. **(C)** Differences in nucleosome occupancy between S2 cells and cells expressing either PH-ML (left) or PH-WT (right) around PH-binding sites (±5 kb). PH-binding sites were aligned by their centers and difference in nucleosome occupancy relative to S2 cells on either side is plotted. **(D)** Differences in nucleosome occupancy between S2 cells and cells expressing either PH-ML (left) or PH-WT (right) ± 5 kb of 6,000 randomly selected non-PH bound sites. Difference in occupancy was obtained by subtracting occupancy values of S2 cell from PH-ML- or PH-WT–expressing cells. **(E)** Heatmaps depicting differences in nucleosome occupancy between S2 cells and cells expressing either PH-ML or PH-WT around ±5 kb of all PH binding-sites or 6,000 random non-PH bound sites. **(F, G, H)** Quantitation of differences in occupancy at PH-binding sites and randomly taken sites. Statistical significance was determined by t test. Outliers are not plotted. **(I, J, K, L)** show representative genomic regions showing nucleosome occupancy in PH-ML, PH-WT–expressing cells or S2 cells at and around PH-binding sites.

binding of PH in more detail. We computed the differences in nucleosome occupancy around (±5 kb) PH-binding sites in cells expressing PH-ML or PH-WT relative to S2 cells. Both PH-WT and PH-ML bind to about 6,300 sites across the genome in *Drosophila* S2 cells and there is no significant difference in the level of PH binding between PH-ML- and PH-WT–expressing cells at about 6,000 sites (32). However, about 4% of sites show a decrease in the level of PH

binding in PH-ML–expressing cells as compared with PH-WT–expressing cells. We determined the change in nucleosome occupancy at 6,000 sites (having same levels of PH binding) by subtracting average nucleosome occupancy of S2 cells from average occupancy obtained from PH-ML- or PH-WT–expressing cells. A significant decrease in nucleosome occupancy is observed in PH-ML–expressing cells as compared with S2 cells. In the case of

PH-WT–expressing cells, a slight increase in occupancy is observed (Fig 1C). Similarly, change in nucleosome occupancy computed at randomly taken 6,000 sites across the genome is shown in Fig 1D. Although at randomly selected sites there is also a slight decrease in nucleosome occupancy in PH-ML–expressing cells, the decrease in nucleosome occupancy around PH-binding sites is about fourfold more than that of randomly selected sites and statistically significant. We also analyzed changes in nucleosome occupancy around PH-binding sites for longer flanking regions (±15 kb) and obtained similar results (Fig S3B). The magnitude of change in nucleosome occupancy at individual sites is shown by heatmaps (Figs 1E and S3A). Quantitation of differences in nucleosome occupancy around the PH-binding sites shows that there is a significant difference between nucleosome occupancy in PH-ML and PH-WT–expressing cells (Fig 1F). For PH-ML–S2 comparison, both the upper and the lower quartiles are below zero, implying that values for difference in occupancy around most of the PH-binding sites are less than zero. Quantitative comparison of change in nucleosome occupancy at PH-binding sites with random sites in PH-ML–expressing cells also shows that nucleosome occupancy is significantly (statistically) lower at PH-binding sites (Fig 1G). However, no significant difference in nucleosome occupancy is observed between PH-binding sites and random sites in PH-WT–expressing cells (Fig 1H). Lower nucleosome occupancy of PH-ML–expressing cells at some representative genomic locations is shown in Fig 1 (Fig 1I–L). This analysis shows that, around PH binding sites, the nucleosome occupancy landscape is altered significantly upon perturbation of PH polymerization. Similarly, chromatin contacts showing a decrease of more than 50% were found to be closer to PH-binding sites (32).

We also computed the occupancy profiles obtained at different concentrations of MNase around PH-binding sites; interestingly, we observed clear difference between the center of PH-binding sites and flanking regions towards MNase digestion. At lower concentrations, flanking regions are more sensitive to MNase digestion than the center of the PH-binding sites. With increase in MNase concentration, more and more reads are released from the center of the PH-binding sites (Fig S4). We observe varying differences in occupancy between three cell lines at different MNase concentrations, emphasizing the need of MNase titration rather than use of just one MNase concentration for evaluating nucleosome occupancy. However, we do not observe any regular positioning of nucleosomes around PH-binding sites.

### PH polymerization modulates MNase-accessibility (MACC)

MNase-titration-Seq also yields a parameter called MACC score in addition to nucleosome occupancy. To determine whether changes in chromatin topology result in changes to nucleosome accessibility, we calculated MACC from our MNase-Titration-Seq data. MACC is the slope of the plot of released (nucleosomal) reads versus log of MNase concentration; a positive slope (positive MACC) indicates high accessibility and a negative slope (negative MACC) indicates low accessibility. MACC has been successfully used to characterize different genomic features like enhancers, promoters, chromatin states, and different classes of TADs (37). MACC has also been used to measure widespread changes in nucleosome

accessibility upon induction of transcription by the unfolded protein response (38). We determined the MACC score in every 300-bp bin genome wide in S2 cells and in cells expressing PH-ML or PH-WT. Comparison of MACC scores between S2 cells with those from cells expressing PH-ML shows that a greater number of genomic sites become accessible in PH-ML–expressing cells (Fig 2A and E). On the other hand, cells over expressing PH-WT show a smaller number of accessible sites in comparison to S2 cells (Fig 2B and F). A comparison between cells expressing PH-ML and cells expressing PH-WT show 6,942 sites with differential accessibility between the two conditions (Fig 2C and G). Furthermore, genome-wide correlation of MACC scores among the three cell lines shows that PH-ML–expressing cells are less similar to S2 cells and the correlation decreases further with cells expressing PH-WT (Fig 2D). These data suggest that, upon disruption of PH polymerization, there is an increase in accessibility of chromatin at some sites and upon over expression of PH, which stabilizes chromatin contacts mediated by PH–PH interactions, there is a decrease in accessibility of chromatin. This is further supported by the observation that many more sites (6,942) show differential accessibility when the condition leading to decease in chromatin contacts (PH-ML) is compared with the condition leading to stabilization of contacts (PH-WT). Computing nucleosome occupancy around the sites having significantly different MACC values between any two cell lines reveals that these sites have slightly higher occupancy than neighboring regions and do not belong to nucleosome-free regions. We also observed that in general, the sites from a particular cell line having higher MACC in comparison to another cell line also have slightly higher occupancy values than another cell line (Fig S5).

### Nucleosome occupancy at PH-mediated chromatin contacts identified by 4C-seq

From the above analysis and shown previously (32), it is clear that both chromatin contacts and nucleosome occupancy are decreased when SAM domain–mediated polymerization of PH is disrupted. To understand the relationship between nucleosome occupancy and chromatin topology, we analyzed the occupancy of nucleosomes underlying chromatin contacts mediated by PH polymerization on *Drosophila* Chr3R. By application of 4C-seq, we have previously mapped PH-SAM domain–mediated chromatin contacts in *Drosophila* S2 cells and in cells expressing either PH-ML or PH-WT (32). A decrease in long range contacts was observed between the bait sequence, at AbdB, and the rest of the chromosome in cells expressing polymerization defective PH-ML relative to S2 cells. However, increased long-range contacts occurred in cells expressing PH-WT. Similarly, baits at Ubx and fab6 show a decrease in contacts with the distal region of BX-C in PH-ML–expressing cells (Fig 3A, data not shown). These changes in 4C-seq detected contacts are relative to contacts observed in S2 cells (32). From these 4C-seq data, we identified 50 nonoverlapping regions on Chr3R involved in chromatin contacts mediated by PH–PH interaction by taking all unique contacts from all three 4C-seq experiments but detected in at least two replicates of 4C-seq data sets (Fig 3B).

To determine a quantitative relationship between nucleosome occupancy and PH-mediated chromatin contacts, we determined

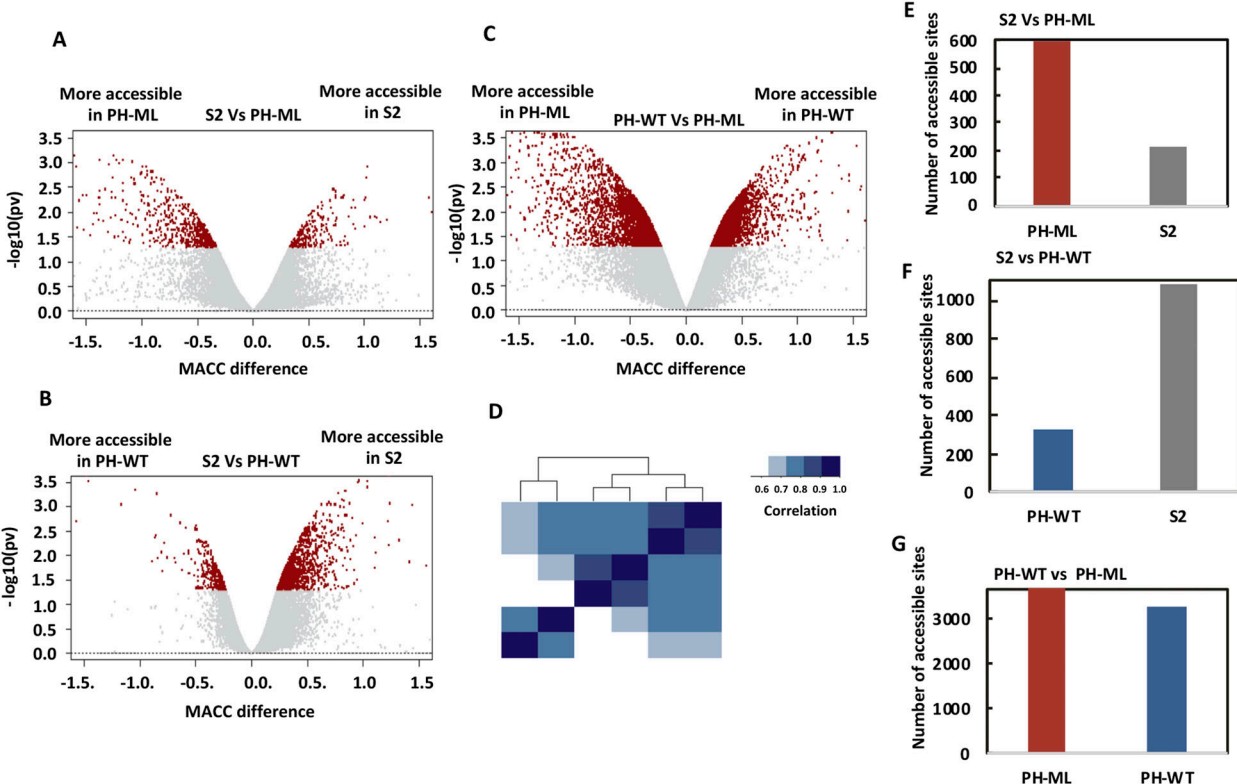

**Figure 2. Comparison of MNase accessibility (MACC).**
**(A, B, C)** show comparison of MACC values between S2 cells versus cells expressing PH-ML, S2 cells versus cells expressing PH-WT and PH-ML versus PH-WT–expressing cells, respectively. Genomic bins (300 bp) showing significant differences in MACC between any two cell lines are shown in red. **(D)** Genome-wide correlation of MACC values between different cell lines and from two different replicates. **(E, F, G)** show the number of more accessible sites form volcano plots in panels (A, B, C), respectively.

the change in nucleosome occupancy at genomic regions corresponding to 50 nonoverlapping PH-mediated contacts on chromosome 3R (Fig 3B). Given the variable genomic size of different contact regions, we binned these regions into 300-bp nonoverlapping bins and compared nucleosome occupancy of about 24,000 300-bp genomic bins underlying these contact regions, shown as a heatmap in Fig 3C. In PH-ML–expressing cells, most of the genomic bins show a decrease in occupancy as compared with corresponding genomic bins in S2 cells (S2 subtracted from PH-ML [PH-ML–S2]). But in PH-WT (PH-WT–S2)–expressing cells, most of the genomic bins showed a slight increase in occupancy in comparison to the corresponding genomic bins from S2 cells. Interestingly, opposite changes in occupancy were observed between many genomic bins from PH-ML- and PH-WT–expressing cells (Fig 3C). On average, a significant decrease in nucleosome occupancy is observed at about 24,000 genomic bins underlying PH-mediated contacts in PH-ML–expressing cells in comparison to S2 cells (Fig 3D). In case of PH-ML–S2 comparison both lower and upper quartiles of the plot are below zero, implying that for most of the genomic regions, the values of difference are less than zero. Hence, PH-ML–expressing cells have lower occupancy than S2 cells at PH-mediated contacts. To further investigate the change in occupancy of nucleosomes because of the change in chromatin topology, we determined changes

in nucleosome occupancy specifically at the contacts that are lost in PH-ML–expressing cells. This analysis was carried for the 4C-seq data set having the bait at Abd-B. A significant difference in nucleosome occupancy is observed between PH-ML- and PH-WT–expressing cells (Fig S6). These data suggest that perturbation of PH SAM domain-mediated chromatin contacts is accompanied by alteration in nucleosome occupancy. However, in comparison to random genomic regions on chr3R, we did not observe significant enrichment of higher MACC sites on regions involved in PH-mediated contacts in PH-ML–expressing cells.

To explore a broader relationship between chromatin topology and nucleosome occupancy, we analyzed the nucleosome occupancy around CTCF-binding sites, many of which are involved in chromatin looping. MNase-seq has been carried out in mouse embryonic stem cells expressing auxin inducible degradable form of CTCF (39). Comparison of nucleosome occupancy around CTCF-binding sites in the presence and absence of auxin analog, indole-3-acetic acid (IAA) shows that nucleosome occupancy decreases when association of CTCF with the chromatin is decreased (Fig S7). To demonstrate the specificity of this change, we determined the change in nucleosome occupancy at 6,000 random non-CTCF-bound sites across the genome and did not observe any significant decrease in nucleosome occupancy (Fig S7). This analysis

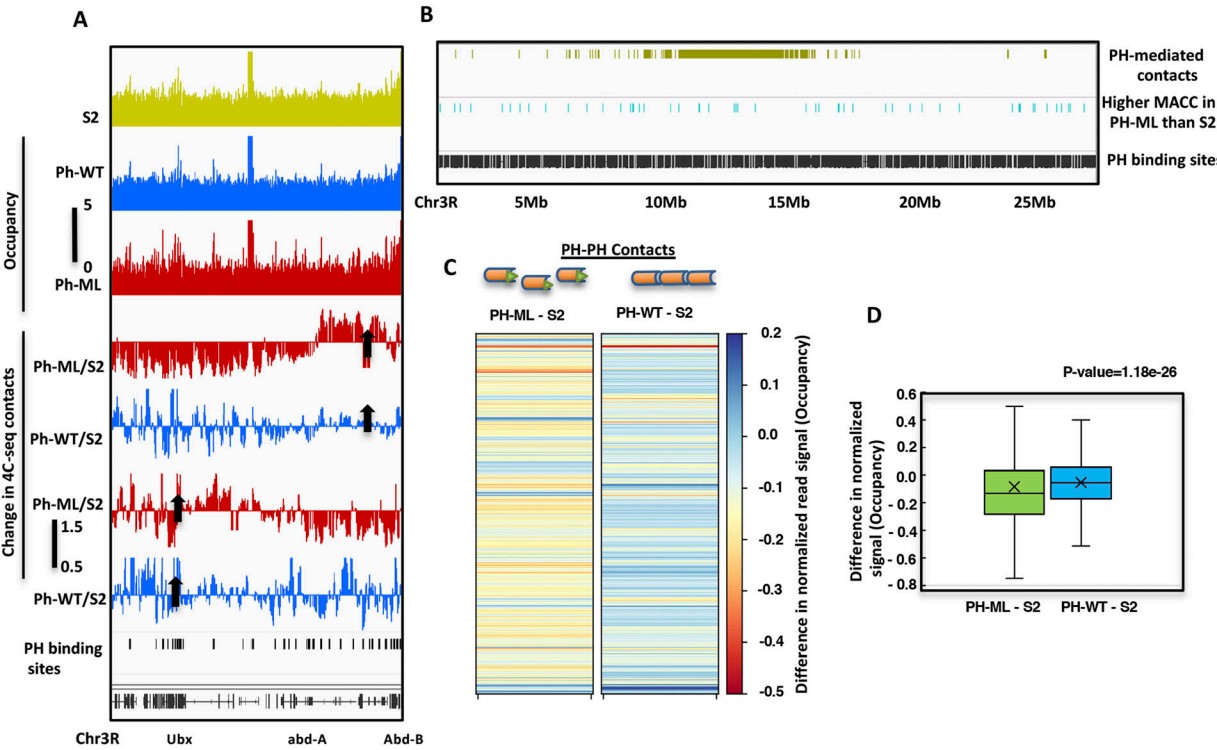

**Figure 3. Nucleosome occupancy at PH-mediated chromatin contacts.**
**(A)** shows genome browser view of nucleosome occupancy for the BX-C in *Drosophila* S2 cells and in cells expressing PH-WT or PH-ML. Change in 4C-seq derived contacts relative to S2 cells in PH-ML- or PH-WT–expressing cells is shown for baits at Abd-B and bxd regions. Arrows show position of baits used in 4C-seq. Black bars show the coordinates of PH-binding sites. **(B)** shows distribution of 50 PH-mediated chromatin contacts on Chr3R and PH-binding sites (black). Cyan bars show regions having higher MACC in PH-ML–expressing cells than in S2 cells. **(C)** shows change in nucleosome occupancy in PH-ML- or PH-WT–expressing cells relative to S2 cells as heatmaps for about 24,000 300-bp genomic bins underlying 50 PH-mediated chromatin contacts. Change in occupancy was obtained by subtracting occupancy values of S2 cells from PH-ML- or PH-WT–expressing cells. **(D)** shows quantitation of change in occupancy in PH-ML- and PH-WT–expressing cells relative to S2 cells for PH-mediated contacts on Chr3R. Outliers are not plotted.

appears to support our results, but a more directed approach is required to delineate the relationship between CTCF-mediated looping and occupancy of underlying nucleosomes.

## Nucleosome occupancy and gene expression

Expression of PH-ML results in change in gene expression (32) (Fig 4), we determined the change in nucleosome occupancy at genes specifically up-regulated in PH-ML, down-regulated in PH-ML or showing no change in gene expression with respect to S2 cells (Fig 4A). Our quantitative analysis shows that nucleosome occupancy is lower in PH-ML–expressing cells than in S2 cells at all three classes of genes irrespective of the change in expression level. As evident from Fig 4, we do not see a decrease in nucleosome occupancy only at up-regulated genes, but most of the genes in all three classes have nucleosome occupancy slightly lower than the corresponding genes from S2 cells. However, we observe a slight decreasing trend in nucleosome occupancy from up-regulated to down-regulated genes (Fig 4B). Previously we have shown that PH is bound to both up-regulated and down-regulated genes (32). Possibly, decrease in nucleosome occupancy can lead to increase in expression at up-regulated genes but indirect effects can also contribute to

change in global gene expression, as a decrease in occupancy is also observed at down-regulated genes.

## Interplay between nucleosome density and chromatin topology

From the above experiments, it is clear that PH polymerization couples nucleosome occupancy and chromatin topology. To understand a mechanistic link between these two properties of chromatin, we used polymer simulations to study the interplay between chromatin contacts and nucleosome density. We employed a minimalistic polymer model that considers chromatin as a bead-spring polymer, with each bead being in a nucleosome bound or dissociated state. A fraction of those beads represent binding sites of non-histone proteins like PH (Fig 5A, red beads). Accounting for the experimentally observed possible coupling between nucleosome occupancy and chromatin contacts, in the model, the chromatin (nucleosome) at the protein-binding sites (red beads) interact with each other via strong specific interactions and all other nucleosome-bound beads (green) interact with each other via weak interactions (see the Materials and Methods section). Simulating this model using Monte Carlo method, we examined the interplay between four different parameters: (a) fraction of specifically interacting beads ($f_{SIB}$) (corresponding to the density of

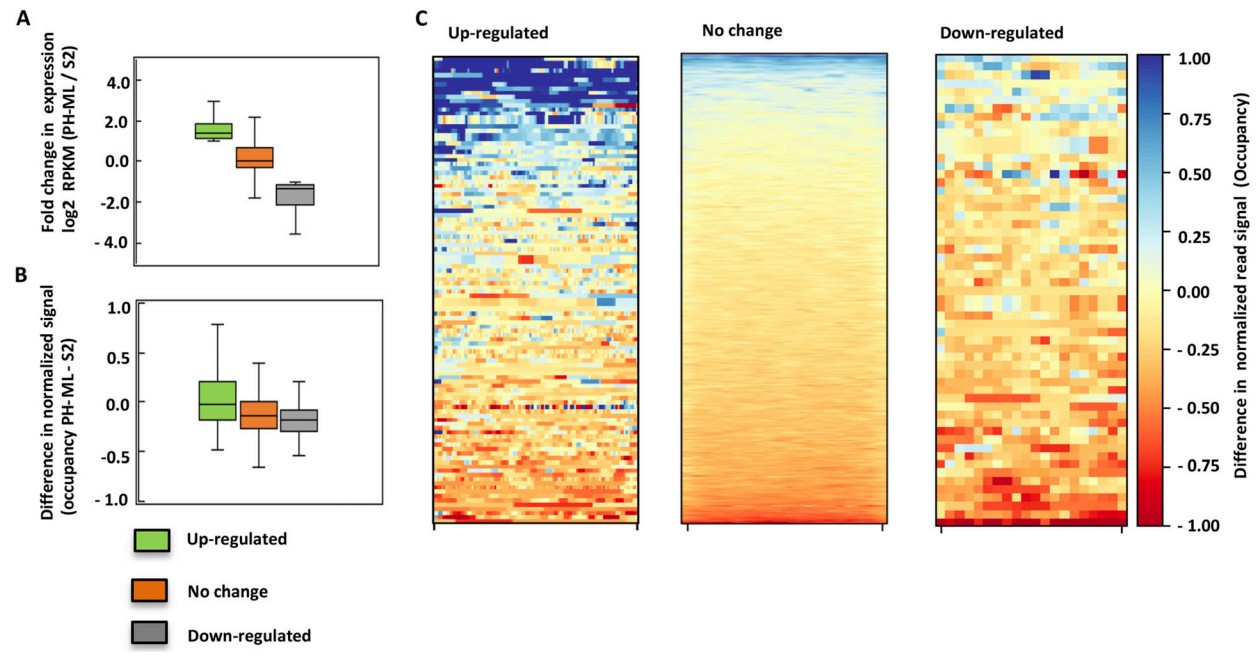

**Figure 4. Nucleosome occupancy and gene expression.**
**(A)** shows log$_2$ fold change in gene expression level between PH-ML–expressing cells and S2 cells. **(B)** shows change in nucleosome occupancy between PH-ML–expressing cells and S2 cells for up- (110) or down (57)-regulated genes and genes showing no change in the expression level (20,586) in PH-ML–expressing cells in comparison to S2 cells. **(C)** Change in nucleosome occupancy at individual genes (gene body) in three different classes of genes is shown as heatmaps.

protein-binding sites), (b) strength of specific interactions ($\varepsilon_S$) between distant red beads (modeling specific protein–protein interactions), (c) weak interaction ($\varepsilon_w$) between nucleosome beads (corresponding to inter-nucleosome interactions, for e.g., interactions between the acidic patch and the H4 histone tail), and (d) nucleosome assembly–disassembly factor (controlled by the parameter $\mu = -\ln(k_{on}/k_{off})$; the higher the value of $\mu$, the more dissociation of nucleosomes). The simulation can answer many interesting questions like how these parameters affect chromatin compaction (radius of gyration $R_g$), and how chromatin contacts influence the nucleosome occupancy.

First, we simulated a N = 200-bead-long polymer having 10% of specifically interacting beads ($f_{SIB}$) (Fig 5B), noting that PH binds approximately to 10% of the sites (PH has 6,300 binding sites with the average size of a PH-binding site being 2.9 kb, in a 143.7-Mb genome). We determined the average nucleosome density ($\overline{\rho}$), for different values of $\mu$ and $\varepsilon_w$, keeping the strength of specific interactions constant. We find that nucleosome density decreases with higher $\mu$ and increases with higher $\varepsilon_w$; examining the $\mu$–$\varepsilon_w$ phase diagram (Fig 5B), we also find two extreme phases—phases of large nucleosome density (light yellow color) and low nucleosome density (blue color) separated by a somewhat sharp boundary (red curve). As we increase the strength of interaction between specifically interacting beads $\varepsilon_S$ = 2–6 k$_B$T, we find that fewer and fewer weak interactions are required to maintain the nucleosome density above 0.5. For example, at $\varepsilon_w$ = 0.4 and $\varepsilon_S$ = 2, even for a very small nucleosome disassembly parameter ($\mu$), the chromatin is in the low nucleosome density phase. As the strength of specific interactions ($\varepsilon_S$) increases, the nucleosome occupancy also increases; the chromatin ends up in a higher nucleosome density state even for

higher $\mu$. In Fig 5C, we present the interplay between nucleosome density and chromatin conformations and how it is influenced by the intra-chromatin interactions and nucleosome disassembly parameter. For $\varepsilon_w$ = 0.4 at lower to moderate values of nucleosome disassembly ($\mu$), the polymer attains a compact conformation with higher density of nucleosomes in the presence of specific interactions ($\varepsilon_S$ = 4) but not in absence of specific interactions ($\varepsilon_S$ = 0). However, upon further increase of nucleosome disassembly, the density of nucleosomes decreases more in unfolded polymers than in folded, showing the interplay between nucleosome disassembly and the specific interactions. We quantified the compaction of the chromatin by determining the radius of gyration ($R_g$) across the landscape of different values for $\varepsilon_S$, $\varepsilon_w$, and $\mu$. Our simulations show that $R_g$ decreases with increase in strength of specific interactions ($\varepsilon_S$) and fewer and fewer weak interactions are required for compaction of the polymer as the value of ($\varepsilon_S$) increases (Fig 5D).

To further evaluate this phenomenon, we kept the strength of specific interactions constant, varied the fraction of specifically interacting beads ($f_{SIB}$ = 5–20%) and determined average nucleosome density ($\overline{\rho}$) at different values of $\mu$ and $\varepsilon_w$ (Figs S8 and S9). We find that increasing the fraction of specifically interacting beads increases the nucleosome occupancy similar to changing the strength of specific interactions (Figs 5, S8, and S9). The polymer compaction also increases with increase in fraction of specifically interacting beads as observed by decrease in values of $R_g$, similar to increase in $\varepsilon_S$.

Given that chromatin is a heteropolymer, to observe a more contrasting effect on nucleosome density because of the presence or absence of distant chromatin interactions, we designed a chromatin polymer of 400 beads with two regions (200 beads each)

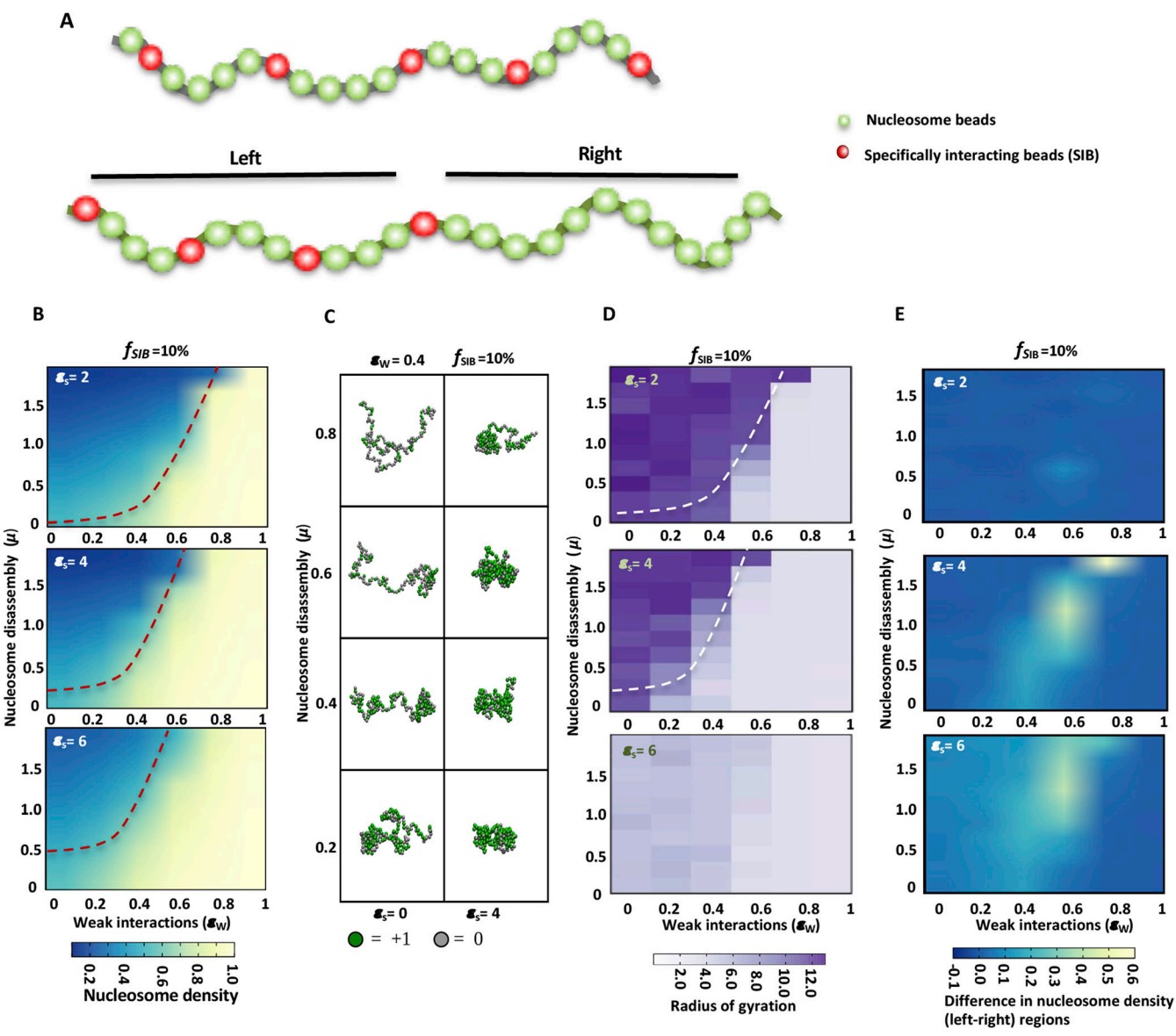

**Figure 5. Nucleosome density and distant chromatin contacts.**
**(A)** Schematic of polymer with specifically interacting beads (SIB) shown as red spheres (above). Schematic of polymer with two regions; left with SIB and right without SIB (below). **(B)** The average nucleosome density is plotted as a heatmap for different values of $\mu$ and $\varepsilon_w$ for polymers having 10% of specifically interacting beads ($f_{SIB}$) but increasing strength of interaction between red beads ($\varepsilon_s$ = 2–6). **(C)** Comparison of polymer conformations with ($\varepsilon_s$ = 4) and without ($\varepsilon_s$ = 0) distant specific interactions at different values of $\mu$ and particular value of ($\varepsilon_w$ = 0.4) and $f_{SIB}$ (10%). **(B, D)** Radius of gyration ($R_g$) values as heatmaps for polymer conformations in the simulations shown in panel (B). **(E)** Difference in nucleosome density (for two-region model) between two regions of polymer (left–right) is plotted as a heatmap for different values of $\mu$ and $\varepsilon_w$ for polymers with increasing values of $\varepsilon_s$ while keeping the fraction of specifically interacting beads ($f_{SIB}$) constant (10%). $\varepsilon_s$ and $\varepsilon_w$ are measured in units of $k_BT$.

as shown in Fig 5A. Here, we introduced specifically interacting beads in one half (left region) but not in the other half (right region) (Fig 5A). After running the simulations, we determined the difference in nucleosome density between two regions. Fig 5E shows the difference in nucleosome occupancy between two regions (left minus right) for various values of $\mu$ and $\varepsilon_w$. We observe an increase in nucleosome occupancy with increase in strength of specific interactions ($\varepsilon_s$ = 2–6k$_B$T) on the left region, which in turn results in an increase in the difference of nucleosome occupancy between the two regions (Fig 5E). Similarly, upon increasing the fraction of specifically interacting beads ($f_{SIB}$ = 5–20%) while keeping the

strength of specific interactions constant, differences in nucleosome occupancy between two regions are visible at intermediate values of $\mu$ and $\varepsilon_w$ (Fig S10B). Fig S10A shows representative snapshots of the chromatin polymer conformation for different values of $\mu$ and $\varepsilon_w$ at constant value of $\varepsilon_s$ and fraction of specifically interacting beads (10%). Here, we observe that for a smaller value of weak interaction strength $\varepsilon_w$, the whole polymer is in an open configuration and has low nucleosome occupancy, whereas for a very high value of $\varepsilon_w$, the whole polymer is in a compact configuration with very high nucleosome occupancy. In both of these cases, there is no difference in the nucleosome occupancy between

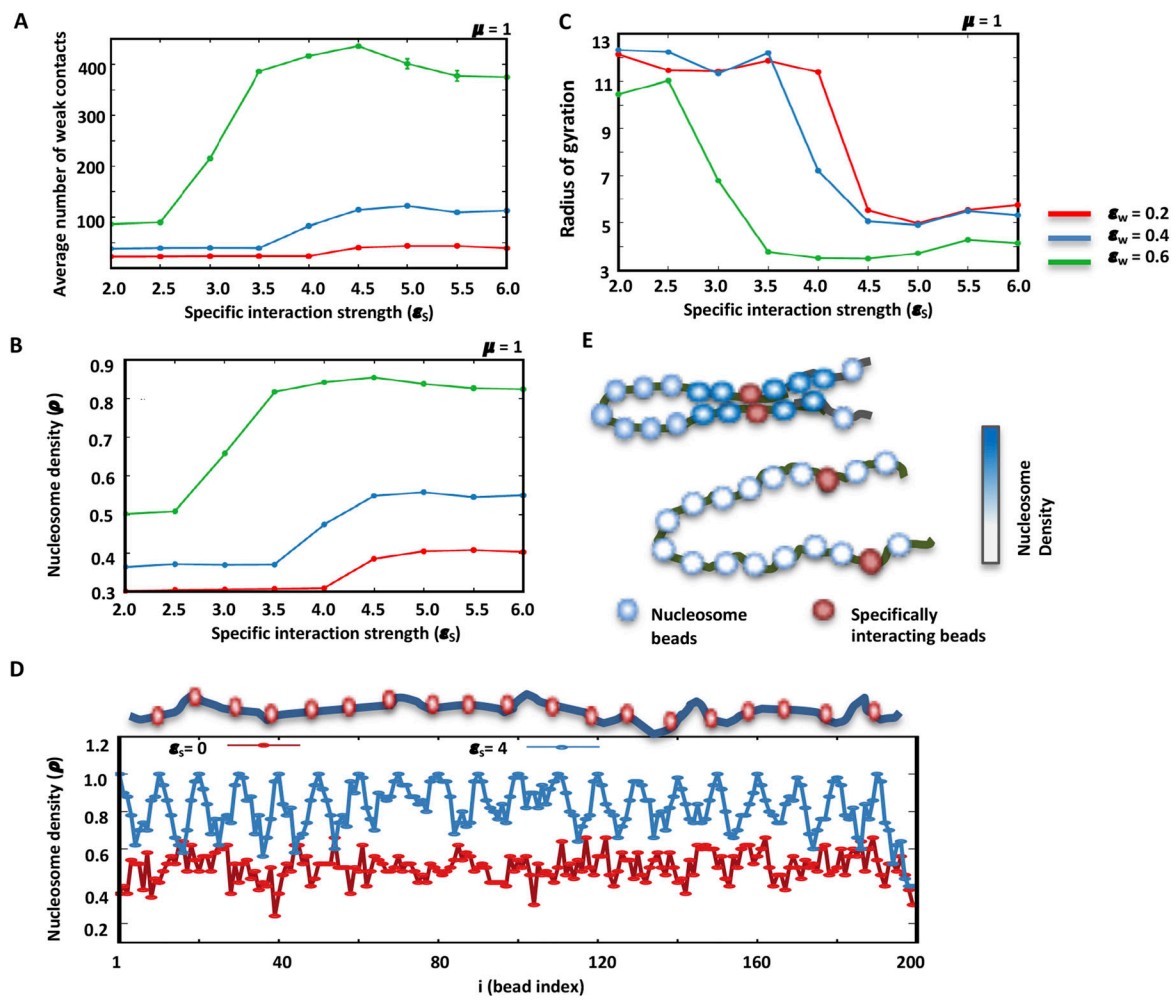

**Figure 6.   Strength of chromatin contacts modulates local weak interactions.**
**(A)** The average number of weak contacts is plotted as a function of strength of specific interaction $\varepsilon_s$ for different values of weak interaction strengths $\varepsilon_w$. The size of the chromatin polymer in these simulations is N = 200 beads with $f_{SIB}$ = 10% and $\mu$ = 1. **(B)** Nucleosome density is plotted as a function of increasing strength of specific interactions ($\varepsilon_s$) at different values of weak interactions ($\varepsilon_w$) and a particular value of $\mu$ = 1 (C) Radius of gyration is plotted as a function of increasing strength of specific interactions ($\varepsilon_s$) at different values of weak interactions ($\varepsilon_w$) and a particular value of $\mu$ = 1 (D) Nucleosome occupancy along the simulated polymer in the presence ($\varepsilon_s$ = 4) or absence ($\varepsilon_s$ = 0) of specific interactions. **(E)** The schematic showing an increase in nucleosome density (occupancy) upon establishment of distant chromatin contact and increase in local inter-nucleosome interactions.

the two regions. On the other hand, we see different higher order structures at intermediate values of $\varepsilon_w$ with optimum value of $\mu$ (for example $\varepsilon_w$ = 0.6 and $\mu$ = 0.8). The left region with specific distant interactions appears folded and having higher nucleosome density than the right region. These simulations lead us to propose that distant chromatin contacts increase the density of underlying nucleosomes.

### Distant chromatin contacts influence weak local interactions

To investigate the possible mechanism for modulation of nucleosome density by strength and density of specifically interacting beads, we analyzed the relationship between specific interactions ($\varepsilon_S$) and weak local interactions ($\varepsilon_w$) (between nucleosomes) in our simulations (with 200-bead polymers). We determined the number of weak local interactions as a function of interaction strength between specifically interacting beads. Fig 6A shows that the

average number of weak interactions increases upon increase in the strength of specific interactions at fixed values of $\mu$ (1.0) and the fraction of specifically interacting beads (10%). This effect is observed at different values of $\varepsilon_w$. As $\varepsilon_w$ increases to 0.6, we see a transition to higher number of average weak contacts even at lower $\varepsilon_S$. With the same parameters, we also estimated the nucleosome density as a function of increasing strength of specific interactions. Fig 6B shows that nucleosome density increases in a manner similar to that of weak contacts (Fig 6A) with increase in the strength of specific interactions, suggesting that distant chromatin contacts lead to increase in local inter-nucleosome interactions, which in turn can lead to stabilization and increase in density of nucleosomes. To find whether this is indeed the case, we analyzed nucleosome occupancy along the polymer (Fig 6D) and found that the nucleosome occupancy is higher around specifically interacting beads when they interact ($\varepsilon_S$ = 4) but not when interaction between them is not established ($\varepsilon_S$ = 0). This agrees with the higher

nucleosome occupancy at PH-binding sites in PH-WT–expressing cells as compared with PH-ML–expressing cells.

The increase in nucleosome density with increase in strength of specific interactions shows a first-order–like transition (Fig 6B) similar to the divalent cation-induced folding transition of chromatin driven by increased local inter-nucleosome interactions (40). To monitor the folding of the polymer, we analyzed the change in $R_g$ with the increase in strength of specific interactions and observed that $R_g$ decreases with increase in $\varepsilon_S$ with a sharp transition (Fig 6C). This suggests that polymer collapses to a compact conformation as a result of specific interactions. Together, these simulation results suggest that distant chromatin contacts can increase nucleosome occupancy, possibly by stabilizing weak local inter-nucleosome interactions (Fig 6E).

## Discussion

In this study, we show that the mutations in the SAM domain of PH which disrupt its polymerization property cause derepression of Hox genes and lead to developmental defects (32, 33, 34), decrease nucleosome occupancy. Decrease in nucleosome occupancy may facilitate changes in gene expression. We have previously shown that the same mutation decreases PH-mediated chromatin contacts. This implies that these two features of chromatin organization (i.e., nucleosome occupancy and chromatin contacts) may be coupled via polymerization of PH. Polymer modeling simulating the interplay between chromatin contacts and nucleosome occupancy suggest that establishing chromatin contacts increases the density of underlying nucleosomes, implying a "top–down" causation property of 3D chromatin organization.

Although hierarchical organization of chromatin has become a text book representation, the biomechanical dependence of different levels of organization on each other has not been studied thoroughly. In a hierarchical organization, different levels have dependences on each other and alterations in one level should get transmitted to other levels lying above or below it (41, 42, 43). In "bottom–up causation" lower levels affect higher levels, whereas in "top–down causation" higher levels of organization can have causal effect on lower levels. In vitro and in silico studies have demonstrated bottom–up causation of chromatin organization by showing that changes in linker length between successive nucleosomes can result in change in folding of the chromatin fiber (44, 45, 46, 47, 48, 49). However, in a hierarchical structure, the formation of higher levels of organization can in turn modulate underlying levels of organization, top–down causation, which has not been explored in 3D chromatin organization. The simulations and experiments presented are consistent with 3D chromatin organization having the property of top–down biomechanical causation. This finding will be important in understanding the principles of chromatin organization and in deciphering the relationship between organization and function.

From previous polymer studies, it seems that establishment of long-range chromatin contacts can exert constraints on the chromatin fiber, decrease its conformational entropy, and facilitate

stabilization of local structures (50, 51, 52, 53, 54, 55, 56). From our simulations and experimentation, this seems likely to be the case. Stronger long-range contacts in our polymer model increase the number of weak local interactions (nucleosome–nucleosome), which stabilize nucleosomes and can result in an increase in nucleosome occupancy. Quantitation of nucleosome occupancy shows a clear decrease at various PH-mediated contact regions on Chr3R including BX-C in PH-ML–expressing cells consistent with model predictions (Fig 3). Consistent with our finding, Isono et al, also proposed that PH polymerization can facilitate an optimum nucleosome density upon establishing noncontiguous chromatin contacts (34). CTCF has been shown to be involved in establishing chromatin loops and according to our simulations, nucleosome occupancy should decrease around CTCF-binding sites upon perturbation of CTCF. Our analysis of the available data set from mouse ES cells (39) shows that nucleosome occupancy decreases upon programed degradation of CTCF. Similarly, in the case of human cells, decrease in occupancy of nucleosomes has also been observed upon depletion of CTCF (57). Distinct positioning of nucleosomes is observed around CTCF-binding sites. However, we do not observe regular positioning of nucleosomes around PH-binding sites. Observation of higher read density (nucleosome occupancy) at the center of PH-binding sites than flanking regions can possibly arise because of very local chromatin compaction activity of a PSC subunit of PRC1. Most of the sites in PH-ML–expressing cells have occupancy lower than corresponding sites from S2, whereas for most of the sites in PH-WT–expressing cells, occupancy is higher than corresponding sites from S2 cells. However, some sites in PH-ML show an increase in occupancy than the corresponding sites from S2 cells (Fig 1E). Examination of these sites revealed that they overlap with genes which are down-regulated in PH-ML–expressing cells than S2 cells. In case of PH-WT–expressing cells, some sites show decrease in occupancy than the corresponding sites from S2 cells and these sites overlap with genes which are up-regulated in PH-WT–expressing cells than S2 cells.

Although we suggest that upon disruption of PH polymerization, the decrease in nucleosome occupancy arises because of disruption of long-range chromatin contacts, but there are other possible indirect effects like changes in levels of histones, global gene expression, and PTMs of histones which in turn can influence the nucleosome occupancy. From RNA-seq analysis, we do not observe any change in the expression level of histones in PH-ML- or PH-WT–expressing cells in comparison to S2 cells. We have also previously shown that the expression of PH-ML does not alter the level of H3K27me3 (32). Recently, it was shown that the expression of PH-ML does not significantly alter H2A-Ub (58). Furthermore, PH-ML (a) forms a PRC1 complex with all subunits as that of PH-WT, (b) binds to the chromatin at same number of sites as that of PH-WT (6,300), and (c) binds to chromatin at a similar level as that of PH-WT (at more than 6,000 sites) (32). We observe decrease in occupancy in PH-ML–expressing cells than S2 cells irrespective of changes in gene expression. Hence, it seems less likely that decrease in nucleosome occupancy observed in PH-ML–expressing cells arises from perturbation of any of these properties of the PRC1 complex or primarily because of change in gene expression or PcG protein-mediated PTMs. Furthermore, the effects observed here are not dramatic but moderate, implying that other

means of chromatin compaction employed by PRC1 are operational (29, 34).

Several lines of evidence support top–down causation by PH–PH–mediated higher order chromatin folding. First multiple studies both in *Drosophila* and mammalian cells using microscopy and chromosome conformation capture-based approaches clearly demonstrate that perturbations of PH lead to loss of chromatin contacts (23, 32, 34, 36, 59, 60, 61). Super-resolution microscopy revealed decompaction of the BX-C in *Drosophila* cells having lower levels of PH (36). In case of *Drosophila* S2 cells, we have shown by 4C-seq that specific mutations which disrupt PH polymerization result in decrease of distal chromatin contacts. Loss of PhC1 in mammalian cells also results in decompaction of polycomb-bound loci and disruption of PhC2 polymerization leads to decondensation of the *Hoxb* locus (34). Decompaction of chromatin can lead to an increase in conformational entropy and, possibly, a decrease in local inter-nucleosome interactions, resulting in decreased occupancy. Recently, the SAM domain of PH was shown to have phase separation property in vitro, condense chromatin (58), and to form nanoscale clusters inside the nucleus (32). The mutations which disrupt PH polymerization weaken but do not prevent phase separation driven by PH-SAM (58) and the same mutations disrupt subnuclear clusters. This suggests that the phase separation property of PH bound to chromatin might contribute to increasing inter-nucleosome interactions and in turn result in stabilization of nucleosomes. However, another possible mechanism that can explain the decrease in nucleosome occupancy is that upon decrease in long-rang chromatin contacts, ATP-dependent chromatin remodeling factors might gain access to the underlying nucleosomes and destabilize them (62, 63). Overall, it seems PRC1 exerts gene repression by multiple means which modulate chromatin from the nucleosome level to higher order organization. PH polymerization seems to contribute to the activity of PRC1 at both these levels. Polymerization of PH may be required to fine-tune the level of nucleosome occupancy required for repression of target loci and a decrease in this level may facilitate binding of factors responsible for gene activation.

We observed significant change in nucleosome occupancy upon perturbation of PH polymerization around PH-binding sites, but no significant change in MACC (accessibility) (data not shown). A previous study in mammalian cells also observed change in nucleosome occupancy around TSS bound by PcG proteins upon deletion of RING1, but no change in accessibility was observed (64). In comparison to PH-binding sites, similar number of random sites across genome also shows a smaller decrease in nucleosome occupancy. Change in nucleosome occupancy and increase in spatial distance of non-PcG sites upon perturbation of PRC1 have been observed previously (19, 64), possibly because of the role of PRC1 as a global regulator of the chromatin structure. We identified many sites which showed differential MACC values across three cell lines (Fig 2). In case of *Drosophila*, the relationship between occupancy and MACC is complex. There are different classes of sites across the *Drosophila* genome based on combination of occupancy and MACC scores: low MACC–low occupancy, high MACC–low occupancy, low MACC–high occupancy, and high MACC–high occupancy (37). The sites having higher MACC values in a cell line (S2/PH-WT/PH-ML) than corresponding sites in another cell line also appear to have slightly higher occupancy.

From previous studies (32, 34, 35, 36, 58) and the results presented here, PH seems to play a very important role in shaping chromatin organization. Its polymerization property mediates chromatin contacts and formation of subnuclear protein clusters. Its phase separation property has also been shown to regulate its compartmentalization and sequestration of chromatin. Taken together, a possible mechanism of chromatin organization seems to evolve from these properties of PH, according to which, PH by its SAM domain polymerization and phase separation properties forms subnuclear compartments which result in chromatin sequestration and establishment of chromatin contacts. The chromatin sequestration and contact formation can increase the local concentration of nucleosomes which facilitates inter-nucleosome interactions resulting in increased occupancy of nucleosomes and gene repression (Fig 7).

# Materials and Methods

## Cell culture

*Drosophila* S2 cells were procured from Expression Systems and were grown at 27°C in ESF921 media. Stable S2 cell lines overexpressing either PH-WT (BLRP-2XFLAG-Ph) or PH-ML (BLRP-2XFLAG-Ph-L1547/H1552R) under inducible metallothionein promoter were grown as described previously by Wani et al (32). Expression of PH in *Drosophila* S2 cells or cell lines expressing PH-WT or PH-ML was checked by immunoblotting using anti-FLAG and anti-PH antibodies (Figs 1 and S1). Given the duplicated nature of ph genes (*ph-p* and *ph-d*) and dominant negative nature of PH-ML mutation, overexpression strategy instead of CRISPR-Cas9 strategy was taken.

## MNase digestion

Cells were grown for three days after induction with 0.5 mM CuSO$_4$ to induce the expression of either PH-WT or PH-ML. Cells were crosslinked in 1% formaldehyde for 10 min at room temperature, tumbling end over end. Crosslinking was quenched by 1 M glycine (from 2.5 M pH 7.9 stock) and cells were tumbled for 10 min at RT. Cells were resuspended in cold PBS (+PI), pelleted through a sucrose cushion (20% sucrose in PBS), resuspended in PBS, and flash frozen in liquid nitrogen at $10^7$ cells per tube and stored at −80°C. For MNase digestion, the cell pellet was resuspended in PBS with 0.1% Triton-X 100 (PBS-TX). Digestion of $10^6$ cells per titration point took place in a volume of 400 $\mu$l PBS-TX supplemented with 1 mM CaCl$_2$. Either 1.5 U, 6.25 U, 25 U, 100 U or 400 U of MNase (Worthington Biochemical) was added to pre-warmed cells and incubated at 37°C for 3 min. Digestion was stopped by moving samples to ice and adding of 10 $\mu$l of 250 mM EDTA, 250 mM EGTA, and 10 $\mu$l of 20% SDS. For DNA cleanup, the digestion products were incubated with RNase (Roche) for 30 min at 37°C, with proteinase K (2.5 $\mu$l of 20 mg/ml) (Roche) for 60 min at 55°C, and incubated at 65°C overnight to reverse crosslinks. After phenol–chloroform extraction and ethanol precipitation, the purified DNA was used as input into the library preparation protocol described in Bowman et al (65), and then

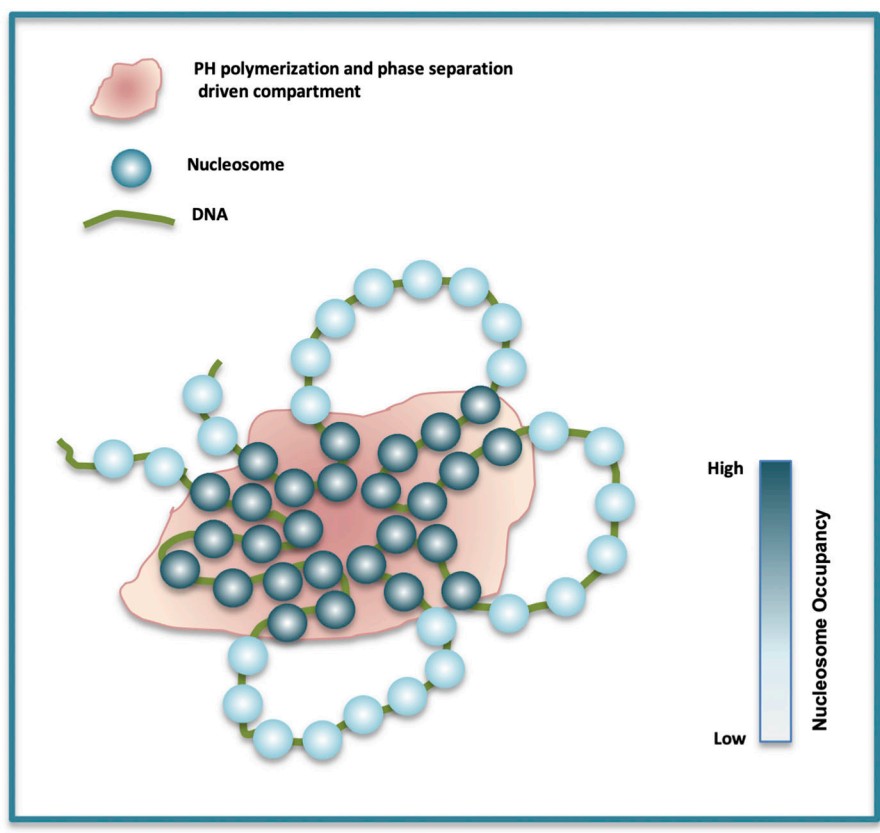

**Figure 7. Possible model of PH-driven chromatin organization.**
PH polymerization and phase separation drives formation of subnuclear compartments leading to sequestration of chromatin and formation of chromatin contacts which increases local nucleosome concentration and inter-nucleosome interactions leading to higher nucleosome occupancy.

sequenced on a HiSeq-2000 sequencer according to the manufacturer's instructions for paired-end sequencing.

### MNase-titration-seq analysis

We sequenced 30 MNase-digested samples and obtained about 1.3 billion paired-end reads. The sequenced paired-end reads obtained at each MNase concentration were mapped to the dm3 version of the *Drosophila melanogaster* genome using Bowtie aligner v.0.12.9. Only reads uniquely mapped with no more than two mismatches were retained. Reads with insert size smaller than 50 bp or greater than 500 bp were removed. As previously, genomic positions with the numbers of mapped tags above the significance threshold (z-score = 7) were discarded (66). Read frequencies were computed in 300-bp non-overlapping bins. For comparisons, read frequencies were normalized by their corresponding library size and represented as reads per million. Normalized read frequencies were plotted around TSS to obtain nucleosome occupancy at different concentrations of MNase.

### Nucleosome occupancy

Averaged nucleosome occupancy from both replicates (with five MNase-seq experiments in each replicate, 2 × 5 = 10 MNase-seq experiments for each cell line) computed over 300 bp bins was used to find differences in nucleosome occupancy across three cell lines using bigwigCompare, computematrix, and plot-heatmap commands from deepTools (67).

### MACC analysis

MACC scores were computed as described previously (37). Briefly, read frequencies were computed in non-overlapping bins of selected size (300 bp) for each titration point independently, normalized to library sizes, and fit with a linear regression. The estimated regression coefficients were corrected to remove dependence on GC content. The corrected values were used as MACC scores.

### Quantitation of change in nucleosome occupancy around PH-binding sites

Coordinates of PH-binding sites were obtained from Wani et al (32) (GSE60686). Averaged nucleosome occupancy from both replicates was obtained around ±5 kb region of each binding site for all three cell lines. All PH-binding sites were aligned by their centers and the changes in nucleosome occupancy on either side were obtained. To obtain changes in nucleosome occupancy around PH-binding sites, averaged nucleosome occupancy (from both replicates) of S2 cells was subtracted from nucleosome occupancy obtained either from PH-ML- or PH-WT–expressing cells and shown as heatmaps (Fig 1). Nucleosome occupancy changes around ±5 kb of PH-binding sites were obtained using the deepTools compute Matrix command (67) and plotted in Fig 1. Random sites (6,000) were taken from the dm3 genome using bedtools (68) and difference in nucleosome occupancy was determined using bigwigCompare command from

deepTools. In Fig S4, averaged read density (from both replicates) at each MNase concentration was computed around PH-binding sites using deepTools compute Matrix command (67).

## Quantitation of change in nucleosome occupancy at gene classes

Gene expression data were obtained from Wani et al (32) GSE72830 and fold change in gene expression level with respect to S2 was calculated for genes specifically up-regulated or down-regulated in PH-ML–expressing cells and genes showing no change in the expression level between PH-ML–expressing cells and S2 cells. Coordinates of gene bodies of all genes were scaled to the same length and the change in nucleosome occupancy was obtained by subtracting average nucleosome occupancy of S2 cells from average nucleosome occupancy of PH-ML–expressing cells. The difference was computed using deepTools and plotted as a heatmap (67).

## Quantitation of change in nucleosome occupancy at PH-mediated chromatin contacts detected by 4C-seq

Coordinates of 4C-seq PH–PH-mediated contacts were obtained from GSE61115. 50 nonoverlapping PH-mediated contacts on chromosome 3R were obtained by taking coordinates of all contacts from all 4C-seq data sets that are present in at least two replicate data sets. Averaged nucleosome occupancy values from two replicate data sets were obtained for 24,000 300 genomic bins spanning 50 PH-mediated chromatin contacts in all three cell lines. Nucleosome occupancy values for corresponding genomic bins from S2 cells were subtracted from those of PH-ML- or PH-WT–expressing cells; changes were plotted as heatmaps (Fig 3).

## Quantitation of change in nucleosome occupancy around CTCF-binding sites

The MNase-seq data set was obtained from GSE131356 and coordinates of the binding sites were obtained from Owens et al (39). Nucleosome occupancies from two experimental replicates were averaged and the difference in occupancy was obtained by subtracting averaged nucleosome occupancy of untreated (−IAA) cells from treated cells (+IAA) using deepTools (68). The difference in nucleosome occupancy was plotted around CTCF sites (±5 kb). Random sites (6,000) were taken from mm9 genome using bedtools (68) and difference in nucleosome occupancy was again plotted (±5 kb). To estimate changes in nucleosome occupancy across the genome, the genome was binned into 3-kb consecutive regions and the difference in occupancy was plotted as a heatmap.

## Statistics and reproducibility

All experiments were carried at least two times. The differences in nucleosome occupancy were computed from the average of two different experiments. Statistical significance for box-whisker plots was checked by $t$ test.

## Polymer modeling

We consider chromatin as a bead-spring polymer consisting of N discrete beads, each of diameter $\sigma$, connected by N − 1 springs. In this coarse-grained model, each bead can be in one of the two states, $\rho = 1$ or $\rho = 0$, representing nucleosome bound state or unbound state, respectively. The total energy of the chromatin polymer, in our model is given by

$$E = V_{sp} + V_{LJ} + \mu \sum_{i=1}^{N} \rho_i - \sum_{(i,j)}^{(3D)} \left[ \varepsilon_s \tau_{ij} \left( \rho_i \times \rho_j \right) + \varepsilon_\omega \left( 1 - \tau_{ij} \right) \left( \rho_i \times \rho_j \right) \right]. \quad (1)$$

The first term represents the spring energy $\sum_{i=1}^{N-1} \frac{1}{2} k_s (q_i - q_0)^2$, where $q_i = |\mathbf{r}_i - \mathbf{r}_{i+1}|$ and $q_0$ is equilibrium length. The second term is the excluded volume term consisting of only repulsive part of the Lennard-Jones potential given by

$$V_{LJ}(r) = \begin{cases} 4\varepsilon \left[ \left( \frac{\sigma}{r} \right)^{12} - \left( \frac{\sigma}{r} \right)^{6} \right] + \varepsilon & r < 2^{1/6}\sigma, \\ 0 & r \geq 2^{1/6}\sigma, \end{cases} \quad (2)$$

The third term represents the chemical potential responsible for nucleosome binding and dissociation such that $\mu = -\ln(k_{on}/k_{off})$ where $k_{on}$ and $k_{off}$ are the rates of nucleosome binding and dissociation, respectively. Hence, large positive $\mu$ implies low nucleosome density. The last term is introduced to study how nucleosome occupancy at any site ($\rho_i$) is connected to the cross-links between different segments resulting in 3D higher order folding. The first part in the last term represents specific interactions between any two sites i and j. If the sites i and j have nucleosomes ($\rho_i = \rho_j = 1$), and have specific interaction ($\tau_{ij} = 1$), then these two sites strongly interact reducing the energy of the system by $\varepsilon_s$. If either of the sites do not have a specific interaction ($\tau_{ij} = 0$), the sites will interact weakly with energy $\varepsilon_w$ as indicated by the second part in the last term. This energy may be considered as weak inter-nucleosome interaction (e.g., between histone tails and acidic patch of nucleosomes). If nucleosomes are not there ($\rho_i$ or $\rho_j = 0$), then none of these interactions will take place. The strong interaction ($\varepsilon_s$) represents the specific protein–protein interactions and the weak interaction ($\varepsilon_w$) represent local inter-nucleosome interactions among regions of chromatin. Note that the summation in this last term of Equation (1) is over the bead pairs i and j which are nearest neighbors in 3D such that the distance $r_{ij}$ between them is less than the cut-off distance $r_{cut} = 1.5\sigma$. We would like to note that binding and dissociation of nucleosomes has some broad similarities with the models having histone modification dynamics (69, 70). We simulate this system using the Metropolis Monte Carlo algorithm. Here, we consider two types of trial moves: (i) in the first trial move, a randomly chosen bead is displaced by a random amount (maximum displacement of 0.5 $\sigma$) along each direction in 3D space; (ii) in the second move, the state $\rho$ of another randomly chosen bead is flipped to the opposite state representing nucleosome binding or disassembly. Each Monte Carlo step consists of N such trial moves of each type. After each trial move, the new configuration is accepted based on the standard Metropolis criteria using the energy given in Equation (1).

### Quantities measured

#### *Average nucleosome density*

We calculated the average nucleosome density of the polymer using

$$\overline{\rho} = \left\langle \frac{1}{N} \sum_{i=1}^{N} \rho_i \right\rangle.$$

The angular brackets denote the ensemble average.

#### *Nucleosome occupancy*

Nucleosome occupancy at any location $i$ is calculated by averaging the nucleosome bead state $\rho_i$ over multiple equilibrium configurations.

#### *Radius of gyration*

The radius of gyration $R_g$ is used to quantify the compactness of the polymer, which is calculated as

$$R_g = \sqrt{\frac{1}{N} \left\langle \sum_{i=1}^{N} (\mathbf{r}_i - \mathbf{r}_{com})^2 \right\rangle}.$$

Here, $\mathbf{r}_{com}$ is the position vector of the center of mass of the polymer.

#### *Number of weak contacts*

We calculated the number of beads which interact with each other through weak interaction $\varepsilon_w$ (the beads having $\rho_i$, $\rho_j$ = 1, and $\tau_{ij}$ = 0).

#### *Parameters*

All the length measurements in the problem are expressed in units of the diameter of a single bead, making $\sigma$ = 1. All the energy scales are expressed in units of thermal energy $k_B T$. The LJ energy parameter is $\varepsilon$ = 1. The constant in spring energy in the first term is $k_s$ = 100 in units of $k_B T/\sigma^2$, with equilibrium length $q_0$ = 1. In the 3D interaction term, beads at a distance $r_{cut}$ < 1.5$\sigma$ are considered as neighbors in 3D.

## Data Availability

The processed and raw data generated in this study have been deposited in the GEO data base under accession number GSE181967. Other data sets used from previously published studies have accession numbers; GSE60686, GSE72830, GSE61115, and GSE131356. The in-house codes are available in GitHub. https://github.com/sangramkadam/LSA-PH-polymerisation-regulates-nucleosome-occupancy.

## Supplementary Information

## Acknowledgements

This work was supported by Wellcome Trust-DBT India Alliance (IA/I/16/2/502713) award to AH Wani. The Department of Biotechnology, UoK is funded by DBT-GoI and DST-FIST from DST-GoI. YA Bhat has senior research fellowship from CSIR-GoI. A Amin is supported by RUSA grant (RUSA-2.2) to UoK. CIRI is supported by PURSE grant from DST-GoI (TPN-56945). S Kadam has senior research fellowship from CSIR-GoI.

### Author Contributions

A Amin: data curation, formal analysis, validation, visualization, methodology, and writing–review and editing.
S Kadam: data curation, software, formal analysis, validation, visualization, methodology, and writing–review and editing.
J Mieczkowski: software, formal analysis, validation, visualization, and methodology.
I Ahmed: software, formal analysis, validation, and visualization.
YA Bhat: formal analysis, methodology, and writing–review and editing.
F Shah: formal analysis, methodology, and writing–review and editing.
MY Tolstorukov: software, formal analysis, validation, visualization, and methodology.
RE Kingston: conceptualization, methodology, and writing–review and editing.
R Padinhateeri: conceptualization, formal analysis, validation, investigation, visualization, and writing–original draft, review, and editing.
AH Wani: conceptualization, formal analysis, supervision, funding acquisition, validation, investigation, methodology, and writing–original draft, review, and editing.

### Conflict of Interest Statement

The authors declare that they have no conflict of interest.

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
