## [Reviewer comments · Life Science Alliance]

Life Science Alliance

Disruption of polyhomeotic polymerization decreases nucleosome occupancy and alters genome accessibility

Adfar Amin, Sangram Kadam, Jakub Mieczkowski, Ikhtlaq Ahmad, Yuonus Bhat, Fouziya Shah, Michael Tolstorukov, Robert Kingston, Ranjith Padinhateeri, and Ajaz Wani

DOI: <https://doi.org/10.26508/lsa.202201768>

Corresponding author(s): Ajaz Wani, University of Kashmir

Review Timeline:

Submission Date:	2022-10-13
Editorial Decision:	2022-11-21
Revision Received:	2023-01-13
Editorial Decision:	2023-02-01
Revision Received:	2023-02-06
Editorial Decision:	2023-02-08
Revision Received:	2023-02-16
Accepted:	2023-02-16

Scientific Editor: Novella Guidi

Transaction Report:

November 21, 2022

Re: Life Science Alliance manuscript #LSA-2022-01768-T

Ajazul H Wani
University of Kashmir

Dear Dr. Wani,

Thank you for submitting your manuscript entitled "Disruption of polyhomeotic polymerization decreases nucleosome occupancy and alters genome accessibility" to Life Science Alliance. The manuscript was assessed by expert reviewers, whose comments are appended to this letter. We invite you to submit a revised manuscript addressing the Reviewer comments.

Thank you for this interesting contribution to Life Science Alliance. We are looking forward to receiving your revised manuscript.

Sincerely,

B. MANUSCRIPT ORGANIZATION AND FORMATTING:

Reviewer #1 (Comments to the Authors (Required)):

Summary

In this manuscript by Amin et al., the authors aimed to understand how long-range chromatin contacts affect nucleosome occupancy. For this, they investigated the effect of polymerization deficient PH [polyhomeotic] protein on nucleosome occupancy and accessibility in *Drosophila* cells (genome-wide and at PH binding sites). PH is sequence specific protein known to induce long range chromatin contacts via PRC1 (Polycomb repressive complex 1) clustering which is important for gene repression. The authors constructed cell lines in which either wild type PH or polymerization deficient PH (PH-ML) is over-expressed using the inducible metallothionein promoter and compared them with the parent S2 cell line. The authors used a quantitative MNase-seq method involving multiple digestion time points which are key to measuring nucleosome occupancy levels and allows measurement of chromatin accessibility. The authors conclude that global nucleosome occupancy and nucleosome occupancy at PH binding sites in PH-ML cells is lower than at random sites or in wild type cells.

The results from simulations to model nucleosome interactions are interesting but I do not have the expertise to comment on them.

I have the following comments for the authors' consideration-

Major Comments:

1. Fig.1 describes the differences in nucleosome occupancy observed in wild type S2 cells, cells over-expressing wild type PH (PH-WT) and cells over-expressing PH-ML. Although it is true that the mean nucleosome occupancy in PH-ML cells is lower relative to wild type cells, the mean difference is only about 0.1%. What is the standard deviation for this mean difference (from replicates)?

The heat map for the range of nucleosome occupancy shows hardly any difference in PH-ML cells when compared with wild type S2 cells (Fig. 1A, left panel). However, the range of nucleosome occupancy differences observed in cells over-expressing PH-WT is much wider (Fig. 1A, right panels). This is true for a comparison of PH-WT with S2 and for PH-WT with PH-ML, suggesting that over-expression of PH-WT alters nucleosome occupancies, whereas PH-ML does not. If this is the case, then the conclusion should be that the altered nucleosome occupancy observed in WT-PH over-expressing cells is dependent on PH polymerization. If true, the claim in the title ("Disruption of polyhomeotic polymerization decreases nucleosome occupancy..") is incorrect. The same argument could be made for nucleosome occupancies at PH binding sites (Fig. 1E). The authors should address this issue.

2. Fig. 2 shows comparisons of MNase accessibility in the same cell lines (MACC analysis). Are these volcano plots primarily detecting nucleosome-free regions (NFRs)? Are the PH binding sites associated with NFRs? And do they change with over-expression?

3. The authors should clarify if Fig. S2 represents nucleosome occupancy data or nucleosome dyad (positioning) data.

4. Fig S7 shows the change in nucleosome occupancy at PH mediated contacts. The total occupancy difference between PH-ML and PH-WT is not great, do the replicates show the same difference? Also, describe how the data is sorted on heat maps?

5. Page 12, Discussion- "We observed a significant change in nucleosome occupancy upon perturbation of PH polymerization around PH binding sites, but no significant change in MACC (accessibility)". Is it possible that in the case of PH-ML cells, the endogenous wild type PH is contributing to polymerization activity at PH sites (before and after the mutant form is expressed). The lack of change in MNase accessibility at PH binding sites in PH-ML cells may be explained by this. What do the authors think about this?

Minor Comments-

1. The legend of Fig 1 is incorrect. (G), (H), (J), (K) are not mentioned.

The y-axis has no label in F, G and H.

2. Fig 4: 'D' is mentioned in the legend but not shown in the figure or found in the text.

3. Page 7: "As shown in figure 5D...". It should be 3D instead of 5D.

4. The title of Figure 4 is confusing.

Reviewer #2 (Comments to the Authors (Required)):

Amin et al. aim to determine how the *Drosophila melanogaster* chromatin-interacting protein Polyhomeotic (PH) influences nucleosome occupancy, in addition to its known roles in Polycomb Repressive Complex 1 and its influence on chromatin

architecture. They perform micrococcal nuclease treatments on previously-established *Drosophila* S2 cells expressing transgenes of PH, either wild-type or a mutant that was previously characterized to disrupt oligomerization. They integrate these new results with data from their previous studies on these cell lines that characterized gene expression and chromatin conformation. They find that nucleosome density generally correlates positively with regions of PH action and oligomerization. Theoretical polymer model simulations support the conclusions that PH-induced increases in long-range chromatin contacts can stabilize weak inter-nucleosome interactions to increase nucleosome density in the 3D region surrounding the PH contact points. The manuscript and its findings have potential to contribute significantly to the field. However, at the moment, some significant questions about the magnitude of the effects and the approach to analysis requires substantial work to more fully support the conclusions and to connect the findings to a clear biological significance.

Major Points:

1. The Y-axis scale for the tracks in Fig. 1I-K bring up an important question: are the difference values reported in Fig. 1A,C-E,F-H directly related to the scale in Fig. 1I-K? If so, these differences are quite small by comparison to the scale of the original values. This is especially true for the mean differences, but extends to differences across individual loci. The scale in Figs. I-K goes from 1.3-4.5, suggesting the background signal is from 0-1.3. The largest changes calculated for the mutant effect in 1E are in this same range, while most are much smaller than this. The authors need to address this issue explicitly to be clear about the scale of effects they observe and how this relates to the biological significance. It may be worth considering a % change analysis for nucleosome occupancy, which may better reflect differences that are seen, rather than using absolute difference.
2. Regarding Point 1, additional statistical analysis for the nucleosome occupancy using replicate comparison should be included in the main text, to solidify the significance of the changes observed.
3. The analysis in Fig. 1E is helpful in isolating distinct populations that can't be appreciated in the metaplots; however the authors do not take advantage of this. For example, what is the significance of the cluster of PH binding sites with reduced nucleosome density in the WT-PH overexpression context?
4. Fig. 3A is out of order in results section and not as easy to interpret before the other panel. Also 3A is very sparsely labeled. What are the genomic coordinates of this region with high density PH-mediated contacts?
5. In Fig. 3D, to better appreciate that the S2 cells have contacts that are lost when PH-ML is expressed, it would help to show a track of S2 cell 4C-Seq.
6. Fig. 3E. While this analysis is useful to consider, having previous data from mouse ESCs in this figure is very confusing. Even with better description of the cell type, this panel should go elsewhere in another figure, if necessary in supplemental figures.
7. Fig. 4C: An analysis of promoter-proximal nucleosomes and PH binding sites of these genes may help sort out why there is higher nucleosome occupancy at some upregulated genes.
8. It may be worth considering how the polymer modeling relates to the biological context, in light of the PH domain sizes being ~3 kb. Does this mean that each bead in the model should be considered a model for roughly 3 kb of chromatin? In light of this scale, it may be worth performing meta-analysis on larger windows in Fig. 1C, for example.

Minor Points

- The authors are encouraged to re-think the color schemes for the genome-wide analysis. Rarely do you see blue denoting higher levels, for example. Also, the color scales change in different analyses, where the "0" mark is not always a consistent color, which is confusing.
- Mention in the Introduction that the bithorax cluster is the Hox gene cluster.
- Shapes symbolizing WT and mutant PH (for example, at the bottom of Fig. 1A panel) need to be used in a clearer manner. For example, the PH-WT v. PH-ML comparison on the right side has a "WT" shape at the bottom, though both WT and mutant are used in the analysis.
- In Fig 2B "WT" should be capitalized.
- In Fig. 5E, adding the cartoon schematic from Fig. S6A would be helpful.

We thank reviewers for giving insightful comments. Incorporating these comments have significantly improved our manuscript.

Reviewer #1 (Comments to the Authors (Required)):

Summary

In this manuscript by Amin et al., the authors aimed to understand how long-range chromatin contacts affect nucleosome occupancy. For this, they investigated the effect of polymerization deficient PH [polyhomeotic] protein on nucleosome occupancy and accessibility in *Drosophila* cells (genome-wide and at PH binding sites). PH is sequence specific protein known to induce long range chromatin contacts via PRC1 (Polycomb repressive complex 1) clustering which is important for gene repression. The authors constructed cell lines in which either wild type PH or polymerization deficient PH (PH-ML) is over-expressed using the inducible metallothionein promoter and compared them with the parent S2 cell line. The authors used a quantitative MNase-seq method involving multiple digestion time points which are key to measuring nucleosome occupancy levels and allows measurement of chromatin accessibility. The authors conclude that global nucleosome occupancy and nucleosome occupancy at PH binding sites in PH-ML cells is lower than at random sites or in wild type cells.

The results from simulations to model nucleosome interactions are interesting but I do not have the expertise to comment on them.

I have the following comments for the authors' consideration-

Major Comments:

1. Fig.1 describes the differences in nucleosome occupancy observed in wild type S2 cells, cells over-expressing wild type PH (PH-WT) and cells over-expressing PH-ML. Although it is true that the mean nucleosome occupancy in PH-ML cells is lower relative to wild type cells, the mean difference is only about 0.1%. What is the standard deviation for this mean difference (from replicates)?

The heat map for the range of nucleosome occupancy shows hardly any difference in PH-ML cells when compared with wild type S2 cells (Fig. 1A, left panel). However, the range of nucleosome occupancy differences observed in cells over-expressing PH-WT is much wider (Fig. 1A, right panels). This is true for a comparison of PH-WT with S2 and for PH-WT with PH-ML, suggesting that over-expression of PH-WT alters nucleosome occupancies, whereas PH-ML does not. If this is the case, then the conclusion should be that the altered nucleosome occupancy observed in WT-PH over-expressing cells is dependent on PH polymerization. If true, the claim in the title ("Disruption of polyhomeotic polymerization decreases nucleosome occupancy..") is incorrect. The same argument could be made for nucleosome occupancies at PH binding sites (Fig. 1E). The authors should address this issue.

i) In Fig.1A the values underlying the heat in PHML-S2 are mostly below zero. A closer look at the heat map key shows that yellow color is below zero although not vary far. Zero values will be little blueish in color. We have now plotted the box and whisker plots for the data in fig1A and find that for PHML-S2 not only the mean but both upper as well as lower quartiles are below zero, suggesting that majority of the genomic regions in PHML expressing cells have nucleosome occupancy less than S2 cells. For PHWT-S2 mean is close to zero, upper quartile as well as a portion of lower quartile are above zero, suggesting that many of the genomic regions in PHWT expressing

cells have occupancy higher than S2 cells. However, the distribution of differences is narrow, implying smaller changes. As expected, the distribution of difference in occupancy between PHWT and PHML (PHWT-PH-ML) cells shifts towards the positive values because PH-ML has lower nucleosome occupancy at most of genomic regions than PHWT expressing cells. The mean differences are -0.0011, 0.0001 and 0.0011 with standard deviations of 0.04, 0.0008 and 0.042, respectively. Here, standard deviation reflects the range of values for differences in nucleosome occupancy between two cell lines.

iii) We have now analyzed the correlation between the differences in nucleosome occupancy from each replicate and as shown below the replicates from a particular cell line are more similar to each other. Furthermore, the correlation analysis of just the occupancy in Fig.1B also shows that replicates from the same cell line are more similar to each other than replicates from other cell lines. We have included this analysis in supplementary figure 2.

iv) In Fig1E, for PHML-S2 majority of the bins have values less than zero as they appear yellow-orange in color. This is clear from the quantitation of heat maps shown in Fig.1F. For PHML-S2, mean, upper quartile as well as lower quartile are below zero suggesting that majority of PH binding sites have nucleosome occupancy lower in PHML expressing cells than in S2 cells.

vi) We have analyzed nucleosome occupancy around PH-binding sites from both the replicates separately and both of them independently show that nucleosome occupancy is lower in PH-ML expressing cells

2. Fig. 2 shows comparisons of MNase accessibility in the same cell lines (MACC analysis). Are these volcano plots primarily detecting nucleosome-free regions (NFRs)? Are the PH binding sites associated with NFRs? And do they change with over-expression?

We thank reviewer for pointing this out. We have now computed the occupancy around the genomic sites showing differential MACC across three cell lines. As shown below they do not seem to correspond to NFRs (as shown below). They appear to have slightly higher occupancy. For example the sites having higher MACC in PH-WT expressing cells than S2 cells have slightly higher occupancy in PHWT expressing cells than S2 cells. Similarly, or PHML expressing cells having higher MACC than S2 cells or PHWT cells have higher occupancy. There are different classes of sites across *Drosophila* genome based on combination of occupancy and MACC scores; low MACC-low occupancy, high MACC-low

occupancy, low MACC-high occupancy and high MACC- high occupancy (Mieczkowski et al., Nature Comm 2016). Here, the differential MACC sites seem to belong to high MACC-high occupancy class of sites.

We have included this analysis as a supplemental figure 5 and explained in the results section.

PH binding sites are not associated with NFR, as shown in Fig.1 I-L and Fig. S5. PH binding sites do not change upon overexpression (Wani et al. Nature Comm 2016).

3. The authors should clarify if Fig. S2 represents nucleosome occupancy data or nucleosome dyad (positioning) data.

We have mentioned in figure legends that it represents nucleosome occupancy and not dyad data.

4. Fig S7 shows the change in nucleosome occupancy at PH mediated contacts. The total occupancy difference between PH-ML and PH-WT is not great, do the replicates show the same difference? Also, describe how the data is sorted on heat maps?

In both the replicates PH-ML has median value lower than PH-WT. (Rep-1 PH-ML – S2 = -0.14, PH-WT – S2 = -0.08; Rep-2 PH-ML – S2 = -0.09, PH-WT – S2 = -0.05)

We agree, although the differences are not large but they are statistically significant. The data in the heat maps have been hierarchically clustered.

5. Page 12, Discussion- "We observed a significant change in nucleosome occupancy upon perturbation of PH polymerization around PH binding sites, but no significant change in MACC (accessibility)". Is it possible that in the case of PH-ML cells, the endogenous wild type PH is contributing to polymerization activity at PH sites (before and after the mutant form is expressed). The lack of change in MNase accessibility at PH binding sites in PH-ML cells may be explained by this. What do the authors think about this?

We agree that endogenous PH is functional in PHML expressing cells. But PHML acts as dominant negative. However, the relationship between nucleosome occupancy and MACC is complex. There are genomic sites which have high occupancy as well as high MACC (high accessibility). There are also sites which have low occupancy and low MACC (Low accessibility) ((Mieczkowski et al., Nature Comm 2016). It is possible that some additional changes/factors may be required around PH binding sites to alter the accessibility.

Minor Comments-

1. The legend of Fig 1 is incorrect. (G), (H), (J), (K) are not mentioned.

The y-axis has no label in F, G and H.

We have now corrected the figure legends and labelled the y-axis in F, G and H.

2. Fig 4: 'D' is mentioned in the legend but not shown in the figure or found in the text.

We have now corrected it.

3. Page 7: "As shown in figure 5D...". It should be 3D instead of 5D.

We have correct it now.

4. The title of Figure 4 is confusing.

We have modified it to now to "nucleosome occupancy and gene expression".

Reviewer #2 (Comments to the Authors (Required)):

Amin et al. aim to determine how the *Drosophila melanogaster* chromatin-interacting protein Polyhomeotic (PH) influences nucleosome occupancy, in addition to its known roles in Polycomb Repressive Complex 1 and its influence on chromatin architecture. They perform micrococcal nuclease treatments on previously-established *Drosophila* S2 cells expressing transgenes of PH, either wild-type or a mutant that was previously characterized to disrupt oligomerization. They integrate these new results with data from their previous studies on these cell lines that characterized gene expression and chromatin conformation. They find that nucleosome density generally correlates positively with regions of PH action and oligomerization. Theoretical polymer model simulations support the conclusions that PH-induced increases in long-range chromatin contacts can stabilize weak inter-nucleosome interactions to increase nucleosome density in the 3D region surrounding the PH contact points. The manuscript and its findings have potential to contribute significantly to the field. However, at the moment, some significant questions about the magnitude of the effects and the approach to analysis requires substantial work to more fully support the conclusions and to connect the findings to a clear biological significance.

Major Points:

1. The Y-axis scale for the tracks in Fig. 1I-K bring up an important question: are the difference values reported in Fig. 1A,C-E,F-H directly related to the scale in Fig. 1I-K? If so, these differences are quite small by comparison to the scale of the original values. This is especially true for the mean differences, but extends to differences across individual loci. The scale in Figs. I-K goes from 1.3-4.5, suggesting the background signal is from 0-1.3. The largest changes calculated for the mutant effect in 1E are in this same range, while most are much smaller than this. The authors need to address this issue explicitly to be clear about the scale of effects they observe and how this relates to the biological significance. It may be worth considering a % change analysis for nucleosome occupancy, which may better reflect differences that are seen, rather than using absolute difference.

i) The scale in panels I-L is not directly related to the scale in panels A, C-G. The scale of 1.3 to 4.5 was arbitrarily taken to highlight changes in nucleosome occupancy. We have now included the full y-axis scale and little more zoomed-in tracks for panels I-L.

ii) We performed an independent quantitative analysis to find changes in nucleosome occupancy across three cell lines. By plotting the volcano plots between fold change and p-value, we identified the 300bp bins having occupancy significantly different across three cell lines. This analysis also shows that there are more sites with decreased occupancy in PH-ML expressing cells than in S2 or PH-WT expressing cells. Therefore, from both analyses shown in Fig1 as well as below same conclusion can be made.

There are about 6000 PH binding sites having an average size of 2.9kb which account for about 10% of genome (143Mb). Given that the change in occupancy is driven by PH polymerization and is more around PH binding sites, a dramatic change in genome-wide nucleosome landscape is not expected. The regions with altered nucleosome occupancy form 9.3, 2.0, and 2.9 percent of altered genome bound by PH in three comparisons PH-WT/PH-ML, PHWT/S2 and PH-ML/S2, respectively. However, these estimates depend upon the thresholds applied in volcano plot. The above-mentioned percentages of genome showing altered nucleosome occupancy were obtained at a conservative threshold of at least 2fold change. A lower threshold of for example 1.5 fold change will increase these percentages.

iii) We agree that observed changes in nucleosome occupancy upon perturbation of PH polymerization are moderate as we have mentioned in discussion, but we show that they are reproducible and statistically significant (Fig. 1, Fig. S2 and S4). Each occupancy value shown is an average of ten different independent MNase-seq experiments for each cell line. Each MNase titration-seq experiment involves digestion of chromatin at five different MNase concs. followed by sequencing and then averaging. Given that two independent biological replicates have been performed, it

leaves us with 2x5 MNase-seq data sets for each cell line, making very unlikely that the differences we see arise just randomly. At genome wide scale the magnitude of differences becomes less but it is much more around PH binding sites. Furthermore, moderate changes in nucleosome occupancy upon perturbation of PRC1 complex in mouse ES have been observed (King et al., Genome Research doi/10.1101/gr.237180.118)

2. Regarding Point 1, additional statistical analysis for the nucleosome occupancy using replicate comparison should be included in the main text, to solidify the significance of the changes observed.

We have now included the comparison of differences in nucleosome occupancy between different cell lines from individual replicates as supplementary figure (Fig. S2) and discussed it in the results also.

3. The analysis in Fig. 1E is helpful in isolating distinct populations that can't be appreciated in the metaplots; however the authors do not take advantage of this. For example, what is the significance of the cluster of PH binding sites with reduced nucleosome density in the WT-PH overexpression context?

We thank reviewer for pointing this out. We checked for possible reason why a fraction of sites show decrease in occupancy in PHWT expressing cells or increase in occupancy around a fraction of sites in PHML expressing cells. We found that the sites around which nucleosome occupancy decreases in PHWT expressing cells most of them overlap with genes which are up-regulated in PHWT cells than S2 cells. Similarly, we observed that the sites around which nucleosome occupancy increases in PHML expressing cells overlap with genes which are down-regulated in PHML expressing cells than S2 cells. We now explain it in discussion on page No. 12.

4. Fig. 3A is out of order in results section and not as easy to interpret before the other panel. Also 3A is very sparsely labeled. What are the genomic coordinates of this region with high density PH-mediated contacts?

We have added more genomic coordinates to the Fig. 3B (corresponding to fig. 3A). We have also rearranged the panels in the figure 3 in the order they appear in the text.

5. In Fig. 3D, to better appreciate that the S2 cells have contacts that are lost when PH-ML is expressed, it would help to show a track of S2 cell 4C-Seq.

The traces show change in contacts in PHML or PHWT expressing cells relative to the S2 cells that is why there is no genome browser trace corresponding to S2 cells itself. We have made it more clear by labelling the traces as PHML/S2 and PHWT/S2. However, traces corresponding to absolute contacts in all three cell lines have been reported earlier (Wani et al. Nature Comm 2016)

6. Fig. 3E. While this analysis is useful to consider, having previous data from mouse ESCs in this figure is very confusing. Even with better description of the cell type, this panel should go elsewhere in another figure, if necessary in supplemental figures. We have now moved the Fig. 3E to the supplementary figures as Fig.S6.

7. Fig. 4C: An analysis of promoter-proximal nucleosomes and PH binding sites of these genes may help sort out why there is higher nucleosome occupancy at some upregulated genes. Upon further analysis of this data, we found there only 17 sites which show increase in occupancy in PH-ML cells (> 0.5) and only 10 of them overlap with PH binding sites. Computing nucleosome occupancy around them did not lead to any conclusion.

8. It may be worth considering how the polymer modeling relates to the biological context, in light of the PH domain sizes being ~ 3 kb. Does this mean that each bead in the model should be considered a model for roughly 3 kb of chromatin? In light of this scale, it may be worth performing meta-analysis on larger windows in Fig. 1C, for example.

We carried the meta-analysis around PH binding sites for longer-flanking distances (± 15 kb), but the results look similar to the analysis carried in Fig1. C. The analysis is shown below and mentioned in the results. We have included this as supplementary figure 3.

Minor Points

- The authors are encouraged to re-think the color schemes for the genome-wide analysis. Rarely do you see blue denoting higher levels, for example. Also, the color scales change in different analyses, where the "0" mark is not always a consistent colour, which is confusing.

We agree with the reviewer that blue is not usually used for increase, however, we want to highlight the decrease in occupancy and for that reason we switched the colours. Given that the magnitude of change is different in different analysis that is why the colour corresponding to zero changes. Keeping the same colour/scale will not clearly reflect changes in some analysis.

- Mention in the Introduction that the bithorax cluster is the Hox gene cluster.

We have now replaced "Hox" with "BX-C"

- Shapes symbolizing WT and mutant PH (for example, at the bottom of Fig. 1A panel) need to be used in a clearer manner. For example, the PH-WT v. PH-ML comparison on the right side has a "WT" shape at the bottom, though both WT and mutant are used in the analysis.

We agree that in case of PHWT-PHML the cartoon is confusing. We have now removed the cartoon from Fig1.A.

- In Fig 2B "WT" should be capitalized.

We have used the capital letters now.

- In Fig. 5E, adding the cartoon schematic from Fig. S6A would be helpful.

We now show the cartoon from Fig. S6A in Fig5A.

February 1, 2023

Re: Life Science Alliance manuscript #LSA-2022-01768-TR

Dr. AJAZ WANI
University of Kashmir
Hazratbal
Srinagar 190006
India

Dear Dr. Wani,

Thank you for submitting your revised manuscript entitled "Disruption of polyhomeotic polymerization decreases nucleosome occupancy and alters genome accessibility" to Life Science Alliance. The manuscript has been seen by the original reviewers whose comments are appended below. While the reviewers continue to be overall positive about the work in terms of its suitability for Life Science Alliance, some important issues remain.

Our general policy is that papers are considered through only one revision cycle; however, given that the suggested changes are relatively minor, we are open to one additional short round of revision. Please note that I will expect to make a final decision without additional reviewer input upon resubmission.

Please submit the final revision within one month, along with a letter that includes a point by point response to the remaining reviewer comments.

To upload the revised version of your manuscript, please log in to your account: <https://lsa.msubmit.net/cgi-bin/main.plex>
You will be guided to complete the submission of your revised manuscript and to fill in all necessary information.

B. MANUSCRIPT ORGANIZATION AND FORMATTING:

Sincerely,

Reviewer #1 (Comments to the Authors (Required)):

The revised version of the manuscript is much improved; however, I have the following comments for the author's consideration.
1) I am not completely convinced that there is much difference in nucleosome occupancy genome wide in PH-ML cells

compared to PH-WT cells based on median and mean values close to 0 (I think PH-WT and PH-ML is correct comparison since both are overexpressed). The authors should include the box plots from both the replicate experiments separately in the main figure 1. What is the unit of occupancy in all the figures, please mention on the Y axis.

2) Unlike the global changes, the difference in nucleosome occupancy at PH binding sites is more convincing. It is important to add this data from both replicates in supplementary figures.

3) Fig. S2 - Are you sure that the plots represent nucleosome occupancy but not the nucleosome center? also, add units at the x-axis (kb?).

Reviewer #2 (Comments to the Authors (Required)):

The revised version by Amin et al has addressed this reviewer's comments in a reasonable manner, for the most part. They provided sufficient evidence that the nucleosome occupancy changes observed were generated in a rigorous way to determine statistical significance. Of note, the authors agreed that these changes are in fact quite moderate across most of the PH binding sites (point iii in rebuttal to Major Point 1). Since the manuscript still needs minor revisions related to figures (see below), I would highly encourage some additional discussion and analysis added. For one, a more clear discussion of the biological significance of such moderate changes in nucleosome occupancy-how does this directly relate to PRC1 function and gene repression? Two, perhaps a direct gene-by-gene correlation analysis of the previously-collected RNA-Seq data and the nucleosome occupancy data, rather than the binning in Fig. 4, would demonstrate that the magnitude of nucleosome occupancy change is in line with the magnitude of gene expression change. Three, the added analysis in supplementary figure S3 suggests that the full extent of the domain that is affected by PH mutation has not been determined. Even a 30 kb metagene region displays the modest, consistent drop in nucleosome occupancy across that entire window, no increases observed toward the edges. How far does this extend, since this drop is not seen at this level for the random sites?

Minor Points:

1. Figure 2 has lost all panel labels (A-G), Figure 3 has lost some panel labels. This is disappointing to see these mistakes in a revised manuscript.

2. It is highly encouraged for Figure 4 analysis to include labels for how many loci are represented in each box plot/heat map.

Reviewer #1 (Comments to the Authors (Required)):

The revised version of the manuscript is much improved; however, I have the following comments for the author's consideration.

1) I am not completely convinced that there is much difference in nucleosome occupancy genome wide in PH-ML cells compared to PH-WT cells based on median and mean values close to 0 (I think PH-WT and PH-ML is correct comparison since both are overexpressed). The authors should include the box plots from both the replicate experiments separately in the main figure 1. What is the unit of occupancy in all the figures, please mention on the Y axis.

We have shown by an independent analysis using volcano plots (in the previous response to point No.1 of reviewer-2) that there are more than 5000 sites which have higher occupancy in PW-WT cells than PH-ML cells. There are about 6000 PH binding sites having an average size of 2.9 kb which account for only about 10% of genome (143Mb), a dramatic change in genome-wide nucleosome landscape is not expected. We include the box plot for replicates in figure S2 which deals with data from replicates. We now mention "normalized read signal" as unit of occupancy

2) Unlike the global changes, the difference in nucleosome occupancy at PH binding sites is more convincing. It is important to add this data from both replicates in supplementary figures.

We have included the data in Fig. S3 now

3) Fig. S2 - Are you sure that the plots represent nucleosome occupancy but not the nucleosome center? also, add units at the x-axis (kb?).

Yes. We have added the units on x-axis.

Reviewer #2 (Comments to the Authors (Required)):

The revised version by Amin et al has addressed this reviewer's comments in a reasonable manner, for the most part. They provided sufficient evidence that the nucleosome occupancy changes observed were generated in a rigorous way to determine statistical significance. Of note, the authors agreed that these changes are in fact quite moderate across most of the PH binding sites (point iii in rebuttal to Major Point 1). Since the manuscript still needs minor revisions related to figures (see below), I would highly encourage some additional discussion and analysis added. For one, a more clear discussion of the biological significance of such moderate changes in nucleosome occupancy-how does this directly relate to PRC1 function and gene repression? Two, perhaps a direct gene-by-gene correlation analysis of the previously-collected RNA-Seq data and the nucleosome occupancy data, rather than the binning in Fig. 4, would demonstrate that the magnitude of nucleosome occupancy change is in line with the magnitude of gene expression change. Three, the added analysis in supplementary figure S3 suggests that the full extent of the domain that is affected by PH mutation has not be determined. Even a 30 kb metagene region displays the modest, consistent drop in nucleosome occupancy across that entire window, no increases observed toward the edges. How far does this extend, since this drop is not seen at this level for the random sites?

We now discuss possible significance of change in nucleosome occupancy in chromatin organization and gene expression by PRC1 (Last few lines on page No.13). As we do not observe a direct correlation between change in nucleosome

occupancy and change in gene expression, gene-by-gene analysis may not yield additional information. We analyzed the meta plots over longer distances (300 kb), we observe that at some places around PH binding sites values go up (close to random) and in case of random sites values come down at many places. Many PH binding sites are in Polycomb domains which spread over varying genomic lengths (10s-100s of kbs), hence, possibly resulting in changes over longer/varying distances.

Minor Points:

1. Figure 2 has lost all panel labels (A-G), Figure 3 has lost some panel labels. This is disappointing to see these mistakes in a revised manuscript.

We have added the labels.

2. It is highly encouraged for Figure 4 analysis to include labels for how many loci are represented in each box plot/heat map.

We mention number of genes for each heat/box plot in figure legends.

February 8, 2023

RE: Life Science Alliance Manuscript #LSA-2022-01768-TRR

Dr. AJAZ WANI
University of Kashmir
Hazratbal
Srinagar 190006
India

Dear Dr. WANI,

Thank you for submitting your revised manuscript entitled "Disruption of polyhomeotic polymerization decreases nucleosome occupancy and alters genome accessibility". We would be happy to publish your paper in Life Science Alliance pending final revisions necessary to meet our formatting guidelines.

- please add a category for your manuscript to our system
- please add the Twitter handle of your host institute/organization as well as your own or/and one of the authors in our system
- please use the [10 author names, et al.] format in your references (i.e. limit the author names to the first 10)
- please add your supplementary figure legends to the main manuscript text
- the Reviewer Token for GSE181967 should be removed from the Data Availability Statement, and the Github codes should be added

Figure Check:

- please add sizes next to blot in Figure S1B

A. FINAL FILES:

B. MANUSCRIPT ORGANIZATION AND FORMATTING:

Sincerely,

February 16, 2023

RE: Life Science Alliance Manuscript #LSA-2022-01768-TRRR

Dr. AJAZ WANI
University of Kashmir
Department of Biotechnology,
University of Kashmir
Hazratbal
Srinagar 190006
India

Dear Dr. WANI,

Thank you for submitting your Research Article entitled "Disruption of polyhomeotic polymerization decreases nucleosome occupancy and alters genome accessibility". It is a pleasure to let you know that your manuscript is now accepted for publication in Life Science Alliance. Congratulations on this interesting work.

DISTRIBUTION OF MATERIALS:

Again, congratulations on a very nice paper. I hope you found the review process to be constructive and are pleased with how the manuscript was handled editorially. We look forward to future exciting submissions from your lab.

Sincerely,
